# Sourcerer: Sample-based Maximum Entropy Source Distribution Estimation

**Julius Vetter**[†,1,2,*]         **Guy Moss**[†,1,2,*]

**Cornelius Schröder**[1,2]        **Richard Gao**[1,2]

**Jakob H. Macke**[1,2,3,*]

[1]Machine Learning in Science, Excellence Cluster Machine Learning, University of Tübingen
[2]Tübingen AI Center
[3]Department Empirical Inference, Max Planck Institute for Intelligent Systems
Tübingen, Germany
[†]Equal contribution.

## Abstract

Scientific modeling applications often require estimating a distribution of parameters consistent with a dataset of observations—an inference task also known as source distribution estimation. This problem can be ill-posed, however, since many different source distributions might produce the same distribution of data-consistent simulations. To make a principled choice among many equally valid sources, we propose an approach which targets the maximum entropy distribution, i.e., prioritizes retaining as much uncertainty as possible. Our method is purely sample-based—leveraging the Sliced-Wasserstein distance to measure the discrepancy between the dataset and simulations—and thus suitable for simulators with intractable likelihoods. We benchmark our method on several tasks, and show that it can recover source distributions with substantially higher entropy than recent source estimation methods, without sacrificing the fidelity of the simulations. Finally, to demonstrate the utility of our approach, we infer source distributions for parameters of the Hodgkin-Huxley model from experimental datasets with hundreds of single-neuron measurements. In summary, we propose a principled method for inferring source distributions of scientific simulator parameters while retaining as much uncertainty as possible.

## 1 Introduction

In many scientific and engineering disciplines, mathematical and computational simulators are used to gain mechanistic insights. A common challenge is to identify parameter settings of such simulators that make their outputs compatible with a set of empirical observations. For example, by finding a distribution of parameters that, when passed through the simulator, produces a distribution of outputs that matches that of the empirical dataset of observations.

Suppose we have a stochastic simulator with input parameters $\theta$ and output $x$, which allows us to generate samples from the forward model $p(x|\theta)$ (which is usually intractable). We have acquired a dataset $\mathcal{D} = \{x_1, ..., x_n\}$ of observations with empirical distribution $p_o(x)$, and want to identify

---

*{firstname.secondname}@uni-tuebingen.de
 Code available at https://github.com/mackelab/sourcerer

38th Conference on Neural Information Processing Systems (NeurIPS 2024).

a distribution $q(\theta)$ over parameters that, once passed through the simulator, yields a "pushforward" distribution of simulations $q^{\#}(x) = \int p(x|\theta)q(\theta)d\theta$ that is indistinguishable from the empirical distribution. This setting is known by different names in different disciplines, for example as *unfolding* in high energy physics [10], *stochastic inverse problems* in various disciplines [7], *population of models* in electrophysiology [30] and *population inference* in gravitational wave astronomy [55]. Adopting the terminology of Vandegar et al. [58], we refer to this task as *source distribution estimation*.

A common approach to source distribution estimation is empirical Bayes [51, 15]. Empirical Bayes uses hierarchical models in which each observation is modeled as arising from different parameters $p(x_i|\theta_i)$. The hyper-parameters of the prior (and thus the source $q_\phi$) are found by optimizing the marginal likelihood $p(D) = \prod_i \int p(x_i|\theta)q_\phi(\theta)d\theta$ over $\phi$. Empirical Bayes has been successfully applied to a range of applications [31, 32, 55]. However, empirical Bayes is typically not applicable to models with intractable likelihoods, which is usually the case for scientific simulators. Using surrogate models for such likelihoods, empirical Bayes has been extended to increasingly more complicated parameterizations $\phi$ of the source distribution, including neural networks [59, 58].

A more general issue, however, is that the source distribution problem can often be ill-posed without the introduction of a hyper-prior or other regularization principles, as also noted in Vandegar et al. [58]: Distinct source distributions $q(\theta)$ can give rise to the same data distribution $q^{\#}(x)$ when pushed through the simulator $p(x|\theta)$ (Fig. 1, illustrative example in Appendix A.7).

We here propose to use the maximum entropy principle, i.e., choosing the "maximum ignorance" distribution within a class of distributions to resolve the ill-posedness of the source distribution problem [19, 24]. The maximum entropy principle formalizes the notion that a good choice for distributions should "assume less". It has been applied to specific source distribution estimation problems in scientific disciplines such as cosmology [23] and high-energy physics [10].

**Our contributions** We introduce *Sourcerer*, a general method for source distribution estimation, providing two key innovations: First, we

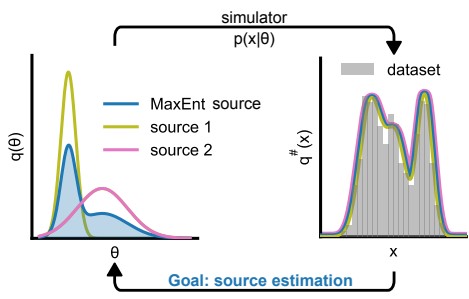

Figure 1: **Maximum entropy source distribution estimation.** Given an observed dataset $\mathcal{D} = \{x_1, \ldots, x_n\}$ from some data distribution $p_o(x)$, the *source distribution estimation* problem is to find the parameter distribution $q(\theta)$ that reproduces $p_o(x)$ when passed through the simulator $p(x|\theta)$, i.e. $q^{\#}(x) = \int p(x|\theta)q(\theta)d\theta = p_o(x)$ for all $x$. This problem can be ill-posed, as there might be more than one distinct source distribution. We resolve this by targeting the maximum entropy distribution, which is unique.

target the maximum entropy source distribution to obtain a well-posed problem, thereby increasing the entropy of the estimated source distributions at no cost to their fidelity. Second, we use general distance metrics between distributions, in particular the Sliced-Wasserstein distance, instead of maximizing the marginal likelihood as in empirical Bayes. This allows evaluation of the objective using *only samples* from differentiable simulators, removing the requirement to have tractable likelihoods. We validate our method on multiple tasks, including tasks with high-dimensional observation space, which are challenging for likelihood-based methods. Finally, we apply our method to estimate the source distribution over the mechanistic parameters of the Hodgkin-Huxley model from a large ($\sim 1000$ samples) dataset of electrophysiological recordings.

## 2 Methods

We formulate the source distribution estimation problem in terms of the maximum entropy principle. The (differential) entropy $H(p)$ of a distribution $p(\theta)$ is defined as

$$H(p) = -\int p(\theta) \log p(\theta)d\theta. \tag{1}$$

## 2.1 Data-consistency and regularized objective

For a given distribution $q(\theta)$ and a simulator with (possibly intractable) likelihood $p(x|\theta)$, the *pushforward* of $q$ is given by $q^{\#}(x) = \int p(x|\theta)q(\theta)d\theta$. The distribution $q(\theta)$ is a source distribution if its pushfoward matches the observed data distribution $p_o(x)$, that is, $q^{\#} = p_o$ almost everywhere. Equivalently, given a distance metric $D(\cdot, \cdot)$ between probability distributions $P(\mathcal{X})$ over the data space $\mathcal{X}$, a source distribution $q$ is one which satisfies $D(q^{\#}, p_o) = 0$. In general, for a given distribution of observations $p_o(x)$ and likelihood $p(x|\theta)$, the source distribution problem is ill-posed as there are possibly many different source distributions. The maximum entropy principle can be employed to resolve this ill-posedness:

**Proposition 2.1.** *Let $Q = \{q|q^{\#} = p_o\}$ be the set of source distributions for a given likelihood $p(x|\theta)$ and data distribution $p_o$. Suppose that $Q$ is non-empty and compact. Then $q^* = \arg\max_{q \in Q} H(q)$ exists and is unique.*

This proposition follows from the fact that the set of source distributions is convex and that the (differential) entropy $H(q)$ is a strictly concave functional. See Appendix A.7 for a proof and additional assumptions.

Proposition 2.1 suggests to solve the constrained optimization problem

$$\max_{\phi} \quad H(q_\phi) \quad \text{s.t.} \quad D(q_\phi^{\#}, p_o) = 0, \quad (2)$$

where $q_\phi$ is some parametric family of distributions.

Practically, however, a solution might not exist, for example due to simulator misspecification. Furthermore, even if a solution exists, it is difficult to obtain since we only have a fixed number of samples from $p_o$ and can thus only estimate $D(q_\phi^{\#}, p_o)$. We therefore propose a *regularized* approximation of Eq. (2) and solve

$$\max_{\phi} \quad \lambda H(q_\phi) - (1-\lambda)\log(D(q_\phi^{\#}, p_o)) \quad (3)$$

instead, where $\lambda$ is a parameter determining the strength of the data-consistency term and the

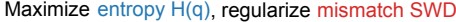

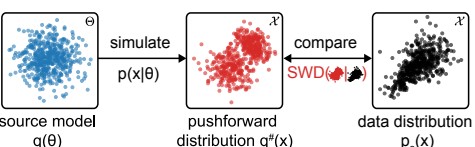

Figure 2: **Overview of Sourcerer.** Given a source distribution $q(\theta)$, we sample $\theta \sim q$ and simulate using $p(x|\theta)$ to obtain samples from the pushforward distribution $q^{\#}(x) = \int p(x|\theta)q(\theta)d\theta$. We maximize the entropy of the source distribution $q(\theta)$ while regularizing with a Sliced-Wasserstein distance (SWD) term between the pushforward of $q^{\#}$ and the data distribution $p_o(x)$ (Eq. (3)). $\Theta$ and $\mathcal{X}$ in top right corner of boxes denote parameter space and data/observation space, respectively.

logarithm is added for numerical stability. This regularized objective is related to the Lagrangian relaxation of Eq. (2), where now $\log D(q^{\#}, p_o) \leq \log \epsilon$ for some $\epsilon > 0$ and the dual variable is $(1-\lambda)/\lambda$.

For $\lambda \to 1$, the loss in Eq. (3) is dominated by the entropy term, and for $\lambda \to 0$ by the data-consistency term. We apply ideas from constrained optimization and reinforcement learning [49, 4, 1] and use a dynamical schedule during training. We initialize training with $\lambda_{t=1} = 1$, and decay this value linearly to a final value $\lambda_{t=T} = \lambda > 0$ over the course of training. This dynamical schedule encourages the variational source model to first explore high-entropy distributions, and later increase consistency with the data between high-entropy distributions. Pseudocode and details of the schedule in Appendix A.3.

## 2.2 Reference distribution

For many tasks, there is an additional constraint in terms of a reference distribution $p(\theta)$. For example, in the Bayesian inference framework, it is common to have a prior distribution $p(\theta)$, encoding existing knowledge about the parameters $\theta$ from previous studies. In such cases, a distribution with higher entropy than $p(\theta)$, even if it is a source distribution, is not always desirable. We therefore adapt our objective function in Eq. (3) to minimize the Kullback-Leibler (KL) divergence between the source $q(\theta)$ and the reference $p(\theta)$:

$$\min_{\phi} \quad \lambda D_{KL}(q||p) + (1-\lambda)\log(D(q^{\#}, p_o)). \quad (4)$$

The KL divergence term can be rewritten as $D_{KL}(q||p) = -H(q) + H(q,p)$, where $H(q,p) = -\int \log(p(\theta))q(\theta)d\theta$ is the cross-entropy between $q$ and $p$. Thus, provided we can evaluate the density $p(\theta)$, we can obtain a sample-based estimate of the loss in Eq. (4). In our work, we consider $p(\theta)$ to be the uniform distribution over some bounded domain $B_\Theta$ (and hence the maximum entropy distribution on this domain). This "box prior" is often used as the naive estimate from literature observations in inference studies. More specifically, in this case, $H(q,p) = -1/|B_\Theta|$, where $|B_\Theta|$ is the volume of $B_\Theta$. Therefore, it is independent of $q$, and hence minimizing the KL divergence is equivalent to maximizing $H(q)$ on $B_\Theta$. In the case where $p(\theta)$ is non-uniform (e.g., Gaussian) the cross-entropy term regularizes the loss by penalizing large $q(\theta)$ when $p(\theta)$ is small.

### 2.3 Sliced-Wasserstein as a distance metric

We are free to choose any distance metric $D(\cdot, \cdot)$ for the loss function Eq. (4). In this work, we use the fast, sample-based, and differentiable Sliced-Wasserstein distance (SWD) [6, 27, 42] of order two. The SWD is defined as the expected value of the one-dimensional Wasserstein distance between the projections of the distribution onto uniformly random directions $u$ on the unit sphere $\mathbb{S}^{d-1}$ in $\mathbb{R}^d$. More precisely, the SWD is defined as

$$\text{SWD}_m(p, q) = \mathbb{E}_{u \sim \mathcal{U}(\mathbb{S}^{d-1})}[W_m(p_u, q_u)], \tag{5}$$

where $p_u$ is the one-dimensional distribution with samples $u^\top x$ for $x \sim p(x)$, and $W_m$ is the one-dimensional Wasserstein distance of order $m$. In the empirical setting, where we are given $n$ samples each from $p_u$ and $q_u$ respectively, the one-dimensional Wasserstein distance is computed from the order statistics as

$$W_m(p_u, q_u) = \left( \sum_{i=1}^{n} ||x_p^{(i)} - x_q^{(i)}||_m^m \right)^{1/m}, \tag{6}$$

where $x_p^{(i)}$ denotes the $i$-th order statistic of the samples from $p_u$ (and similarly for $x_q^{(i)}$), and $|| \cdot ||_m$ denotes the $L^m$ distance on $\mathbb{R}$ [47]. The time complexity of computing the sample-based one-dimensional Wasserstein distance is thus the time complexity of computing the order statistics, which is $\mathcal{O}(n \log n)$ in the number of datapoints $n$ [6]. This is significantly faster than computing the multi-dimensional Wasserstein distance ($\mathcal{O}(n^3)$, 29), or the commonly used Sinkhorn algorithm for approximating the Wasserstein distance ($\mathcal{O}(n^2)$ 47). While the SWD is not the same as the multi-dimensional Wasserstein distance, it is still a valid metric on the space of probability distributions. In particular, the SWD converges quickly with rate $O(\sqrt{n})$ to its true value [41, 42].

### 2.4 Differentiable simulators and surrogates

Our method only requires that sampling from the simulator $p(x|\theta)$ is a differentiable operation. In practice, however, many simulators do not satisfy this property. For such simulators, we first train a surrogate model. In particular, our method can make use of surrogates that model the likelihood only implicitly. Such surrogate models can be easier to train and evaluate in practice. This is a distinct requirement from likelihood-based approaches such as Vandegar et al. [58], which require that the likelihood $p(x|\theta)$ can be evaluated explicitly *and* is differentiable. This means that our sample-based approach can be readily applied to a larger set of simulators than likelihood-based approaches.

### 2.5 Source model and entropy estimation

In this work we use neural samplers as proposed in Vandegar et al. [58] to parameterize a source model $q_\phi$. These samplers employ unconstrained neural network architectures (in our case a multi-layer perceptron) to transform a random sample from $z \in \mathcal{N}(0, I)$ into a sample from $q_\phi$. While neural samplers do not have a tractable likelihood, they are faster to evaluate than models with tractable likelihoods. Furthermore, by using unconstrained network architectures, neural samplers are flexible and additional constraints (e.g., symmetry, monotonicity) are easy to introduce.

To use likelihood-free source parameterizations, we require a purely sample-based estimator for the entropy $H(q_\phi)$. This can be done using the *Kozachenko-Leonenko* entropy estimator [28, 3], which is based on a nearest-neighbor density estimate. We use the Kozachenko-Leonenko estimator in this work for its simplicity, but note that sample-based entropy estimation is an active area of research, and other choices are possible [48]. Details about the Kozachenko-Leonenko estimator can be found in Appendix A.6.

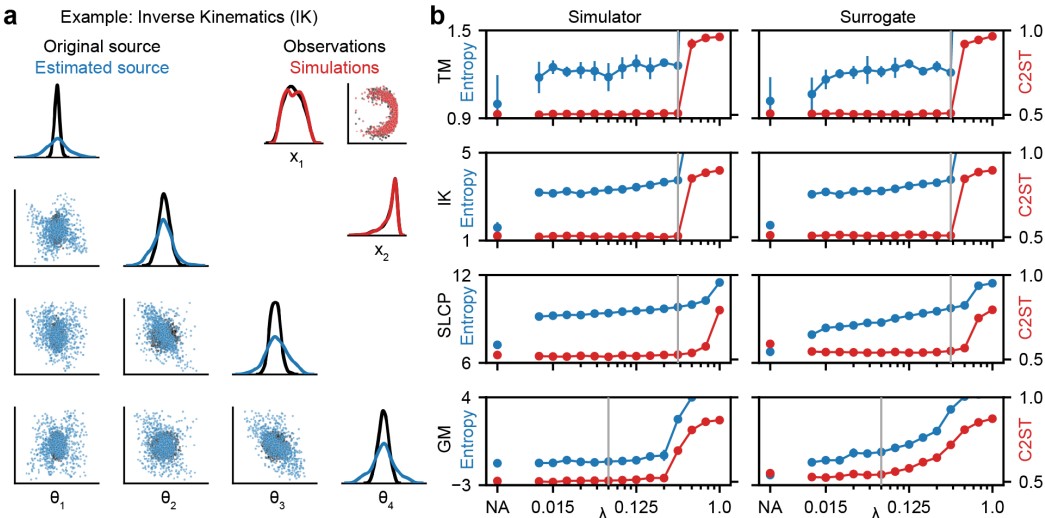

Figure 3: **Results for the source estimation benchmark.** **(a)** Original and estimated source and corresponding pushforward for the differentiable IK simulator ($\lambda = 0.35$). The estimated source has higher entropy than the original source that was used to generate the data. The observations (simulated with parameters from the original source) and simulations (simulated with parameters from the estimated source) match. **(b)** Performance of our approach for all four benchmark tasks (TM, IK, SLCP, GM) using both the original (differentiable) simulators, and learned surrogates. Source estimation is performed without (NA) and with entropy regularization for different choices of $\lambda$. For all cases, mean C2ST accuracy between observations and simulations (lower is better) as well as the mean entropy of estimated sources (higher is better) over five runs are shown together with the standard deviation. The gray line at $\lambda = 0.35$ ($\lambda = 0.062$ for GM) indicates our choice of final $\lambda$ for the numerical benchmark results (Table 1).

## 3 Experiments

To evaluate the data-consistency and entropy of source distributions estimated by Sourcerer, we benchmark our method against Neural Empirical Bayes (NEB) [58], a state-of-the-art approach to source distribution estimation. The benchmark comparison is performed on four source distribution estimation tasks including three presented in Vandegar et al. [58]. We then demonstrate the advantage of Sourcerer in the case of differentiable simulators with a high-dimensional data domain, where likelihood-based empirical Bayes approaches would require training a likelihood surrogate. Finally, we use Sourcerer to estimate the source distribution for a Hodgkin-Huxley simulator of single-neuron voltage dynamics from a large dataset of experimental electrophysiological recordings. For all tasks except the Hodgkin-Huxley task (where the observed dataset is experimentally measured), we generate two datasets of observations of equal size from the same reference source distribution. The first is used to train the source model, and the second is used to evaluate the quality of the learned source.

### 3.1 Source Estimation Benchmark

**Benchmark tasks** The source estimation benchmark contains four simulators: two moons (TM), inverse kinematics (IK), simple likelihood complex posterior (SLCP), and Gaussian Mixture (GM) (details about simulators and source distributions are in Appendix A.2). Notably, all four simulators are differentiable. Therefore, we can evaluate our method directly on the simulator as well as trained surrogates. For all four simulators, source estimation is performed on a synthetic dataset of 10000 observations that were generated by sampling from a pre-defined original source distribution and evaluating the resulting pushforward distribution using the corresponding simulator. The quality of the estimated source distributions is measured using a classifier two sample test (C2ST) [33] between the observations and simulations from the source. We also report the entropy of the estimated sources. Given two sources with the same C2ST accuracy, the higher entropy source is preferable. We compare

Table 1: **Numerical benchmark results for Sourcerer.** We show the mean and standard deviation over five runs for differentiable simulators and surrogates of Sourcerer on the benchmark tasks, and compare to NEB. All approaches achieve C2ST accuracies close to 50%. For the Sliced-Wasserstein-based approach, the entropies of the estimated sources are substantially higher (bold) with the entropy regularization ($\lambda = 0.35$ for TM, IK, SLCP, $\lambda = 0.062$ for GM, gray line in Fig. 3).

| Method | | Sourcerer Sim. (with reg.) | Sourcerer Sim. (w/o reg.) | Sourcerer Sur. (with reg.) | Sourcerer Sur. (w/o reg.) | NEB |
|---|---|---|---|---|---|---|
| TM | C2ST acc. | 0.51 (0.004) | 0.5 (0.008) | 0.51 (0.003) | 0.51 (0.006) | 0.53 (0.005) |
| | Entropy | **1.26** (0.022) | 1.0 (0.198) | **1.21** (0.054) | 1.02 (0.162) | 1.13 (0.093) |
| IK | C2ST acc. | 0.51 (0.002) | 0.51 (0.005) | 0.51 (0.005) | 0.51 (0.01) | 0.6 (0.014) |
| | Entropy | **3.75** (0.066) | 1.59 (0.246) | **3.78** (0.022) | 1.7 (0.165) | 0.82 (0.712) |
| SLCP | C2ST acc. | 0.53 (0.005) | 0.53 (0.006) | 0.55 (0.003) | 0.59 (0.017) | 0.53 (0.006) |
| | Entropy | **9.81** (0.039) | 7.23 (0.052) | **9.74** (0.039) | 6.76 (0.302) | 7.56 (0.097) |
| GM | C2ST acc. | 0.51 (0.005) | 0.5 (0.006) | 0.54 (0.006) | 0.55 (0.005) | 0.52 (0.004) |
| | Entropy | **-1.12** (0.083) | -1.25 (0.106) | **-0.36** (0.095) | -2.19 (0.212) | -1.5 (0.052) |

to the NEB estimator with the same parameterization of the source model and 1024 Monte Carlo samples to estimate the marginal likelihood (details in Appendix A.3).

**Benchmark performance**   We first check whether minimizing the Sliced-Wasserstein distance without any entropy regularization finds good source distributions. This corresponds to the case $\lambda = 0$ in Eq. (3) without any decay. In this way, we compare the data-consistency objective in Eq. (4) to the NEB objective of maximizing the marginal likelihood. We find that for the differentiable simulators, the Sliced-Wasserstein-based approach is able to find good source distributions with C2ST accuracies close to 50% for all benchmark tasks (Fig. 3, labeled NA). This also applies when we use surrogate models to generate the pushforward distributions. In particular, the quality of the estimated source distributions matches those found by NEB (Table 1).

We then apply entropy regularization as defined in Eq. (3) for all benchmark tasks. The entropy of the estimated sources is drastically increased *without* any cost in the quality of the simulations (Fig. 3b). While C2ST accuracy remains close to 50% across all benchmark tasks, the entropy of estimated sources is substantially higher than that of sources estimated with NEB, or when minimizing only the data-consistency term (Table 1). We also explore the dependence of the results on the final regularization strength $\lambda$ (Fig. 3b). We observe a sharp trade-off: above a critical value of $\lambda$, the SWD term becomes too weak, and the fidelity of the simulations rapidly declines. However, below this critical value of $\lambda$, the results are robust relative to $\lambda$: the estimated sources produce simulations that match the observations, and have comparable entropy.

Additionally, for both IK and SLCP simulators, the entropy of the sources estimated by our method is higher than the entropy of the original source distribution (Fig. 3a and Fig. A7) despite the simulations and observations being indistinguishable from each other (C2ST accuracy: 50%). This does not contradict our approach: The original source distribution just happens not to be the maximum entropy source for these simulators.

We also investigate the robustness of our approach to the choice of the differentiable, sample-based distance by repeating all experiments for these benchmark tasks using the Maximum Mean Discrepancy (MMD, 22) and find comparable results (Fig. A4). Finally, we demonstrate (Fig. A5) the robustness of our approach for small dataset sizes by repeating the Two Moons task with ($N = 100$) observations (as opposed to 10000), and for high-dimensional parameter spaces by repeating the Gaussian Mixture task with $D = 25$ dimensions (as opposed to 2).

### 3.2   High-dimensional observations: Lotka-Volterra and SIR

Since our method is sample-based and does not require likelihoods, it is possible to estimate sources by back-propagating through the differentiable simulators directly. This is advantageous especially for simulators with high-dimensional outputs, as we no longer require to first train a surrogate likelihood model, which can be challenging when faced with high-dimensional data such as time series. Here, we

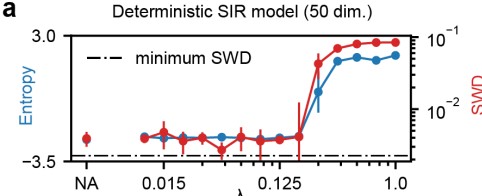
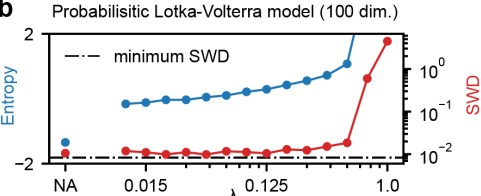

Figure 4: **Source estimation on differentiable simulators.** For both the deterministic SIR model **(a)** and probabilistic Lotka-Volterra model **(b)**, the Sliced-Wasserstein distance (lower is better) between observations and simulations as well as entropy of estimated sources (higher is better) for different choices of $\lambda$ and without the entropy regularization (NA) are shown. Mean and standard deviation are computed over five runs.

highlight this capability of our method by estimating source distributions for two high-dimensional, differentiable simulators: The Lotka-Volterra model and the SIR (Susceptible, Infectious, Recovered) model. The Lotka-Volterra model is used to model the density of two populations, predators and prey. The SIR model is commonly used in epidemiology to model the spread of disease in a population (details about both models and source distributions in Appendix A.2). Compared to the benchmark tasks in Sec. 3.1, the dimensionality of the data space is much larger: Both the Lotka-Volterra and the SIR model are simulated for 50 time points resulting in a 100 and 50 dimensional time series, respectively.

Furthermore, to show that unlike NEB (which maximizes the marginal likelihood), our sample-based approach is applicable to deterministic simulators, we use a deterministic version of the SIR model with no observation noise. Similarly to the benchmark tasks, we define a source, and simulate 10000 observations using samples from this source to define a synthetic dataset on which to perform source distribution estimation. Here, we directly evaluate the quality of the estimated source distributions using the Sliced-Wasserstein distance. We compare this distance to the minimum expected distance, which is the distance between simulations of different sets of samples from the same original source. For a comparison with NEB, we train surrogate models with a reduced dimensionality and again compute C2ST accuracies and entropies of the estimated sources (see Appendix A.5 and Fig. A3 for details on surrogate training and pushforward plots).

**Source estimation for the deterministic SIR model**   Our method is able to estimate a good source distribution for the deterministic SIR model: The Sliced-Wasserstein distance between simulations and observations is close to the minimum expected distance (Fig. 4a). In contrast to the benchmark tasks, estimating sources with entropy regularization does not lead to an increase in entropy for the SIR model, and the quality of the estimated source remains constant for various choices of $\lambda$. A possible explanation for this is that there is no degeneracy in the parameter space of the deterministic simulator, and there exists only one source distribution.

**Source estimation for the probabilistic Lotka-Volterra model**   For the probabilistic Lotka-Volterra model, our method is also capable of estimating source distributions. As for the SIR model, the Sliced-Wasserstein distance between simulations and observations is close to the minimum expected distance (Fig. 4b). However, unlike the SIR model, estimating the source with entropy regularization yields a large increase in entropy compared to when not using the regularization. For the Lotka-Volterra model, our method yields a substantially higher entropy at no additional cost in terms of source quality.

When using the surrogate models with reduced dimensionality to estimate the source distributions, we find that Sourcerer achieves better C2ST accuracies than NEB. Furthermore, for the Lotka-Volterra model, the entropy regularization again leads to a substantial increase in the entropy of the estimated sources (Table 2). In summary, the experiments on the SIR and Lotka-Volterra models show that our approach is able to scale to higher dimensional problems and can use gradients of complex simulators to estimate source distributions directly from a set of observations.

Table 2: **Numerical results for the SIR and Lotka-Volterra model** We show the mean and standard deviation over five runs for differentiable simulators and surrogates of Sourcerer on the high-dimensional SIR and Lotka-Volterra (LV) models, and compare to NEB. For the comparison with NEB, we train the required surrogate models with reduced dimensionality (25 dimensions instead of 50 or 100). Sourcerer achieves C2ST accuracies close to 50%. For NEB, the C2ST accuracies are worse. For the LV model, the entropies of the estimated sources are higher with the entropy regularization ($\lambda = 0.015$ for SIR, $\lambda = 0.125$ for LV).

| Method | | Sourcerer Sim. (with reg.) | Sourcerer Sim. (w/o reg.) | Sourcerer Sur. (with reg.) | Sourcerer Sur. (w/o reg.) | NEB |
|---|---|---|---|---|---|---|
| SIR | C2ST acc. | 0.56 (0.013) | 0.56 (0.015) | 0.55 (0.005) | 0.55 (0.005) | 0.76 (0.024) |
| | Entropy | -2.3 (0.079) | -2.37 (0.169) | -2.29 (0.076) | -2.5 (0.05) | -0.63 (0.174) |
| LV | C2ST acc. | 0.57 (0.009) | 0.52 (0.001) | 0.56 (0.005) | 0.54 (0.009) | 0.62 (0.011) |
| | Entropy | **0.29** (0.017) | -1.34 (0.087) | **0.34** (0.05) | -1.01 (0.13) | -1.28 (0.073) |

### 3.3 Estimating source distributions for a single-compartment Hodgkin-Huxley model

**Single-compartment Hodgkin-Huxley simulator and summary statistics** The single-compartment Hodgkin-Huxley model consists of a system of coupled ordinary differential equations simulating different ion channels in a neuron. We use the simulator described in Bernaerts et al. [2] with 13 parameters. In data space, we use five commonly used summary statistics of the observed and simulated spike trains. These are the (log of the) number of spikes, the mean of the resting potential, and the mean, variance and skewness of the voltage during external current stimulation. As the internal noise in the simulator has little effect on the summary statistics, we train a simple multi-layer perceptron as surrogate on $10^6$ simulations. The parameters used to generate these training simulations were sampled from a uniform distribution that was used as the prior in Bernaerts et al. [2] (details on simulator, choice of surrogate and the surrogate training in Appendix A.9).

Using this surrogate, we estimate source distributions from a real-world dataset of electrophysiological recordings. The dataset [52] consists of 1033 electrophysiological recordings from the mouse motor cortex. In general, parameter inference for Hodgkin-Huxley models can be challenging as models are often misspecified [56, 2]. Thus, estimating the source distribution for this task is useful for downstream inference tasks, as the prior knowledge gained can significantly constrain the parameters of interest.

**Source estimation for the Hodgkin-Huxley model** On visual inspection, simulations from the estimated source look similar to the original recordings (all observations spike at least once, spikes have similar magnitudes) and show none of the unrealistic properties (e.g., spiking before the stimulus is applied) that can be observed in some of the box uniform prior simulations (Fig. 5a). This match is also confirmed by the distribution of summary statistics, which match closely between simulations and observations (Fig. 5b). Furthermore, our method achieves good C2ST accuracy of $\approx 61\%$ for different choices of $\lambda$ (Fig. 5d), as well as a small Sliced-Wasserstein distance of $\approx 0.08$ in the standardized space of summary statistics (Fig. 5e). While the source estimated without entropy regularization also achieves good fidelity, its entropy is significantly lower than any of the source distributions estimated with entropy regularization (Fig. 5d/e, example source distribution in Fig. 5c, full source in Fig. A11).

Overall, these results demonstrate the importance of estimating source distributions using the entropy regularization, especially on real-world datasets: Estimating the source distribution without any entropy regularization can introduce severe bias, since the estimated source may ignore entire regions of the parameter space. In this example, the parameter space of the single-compartment Hodgkin-Huxley model is known to be highly degenerate, and a given observation can be generated by multiple parameter configurations [14, 39].

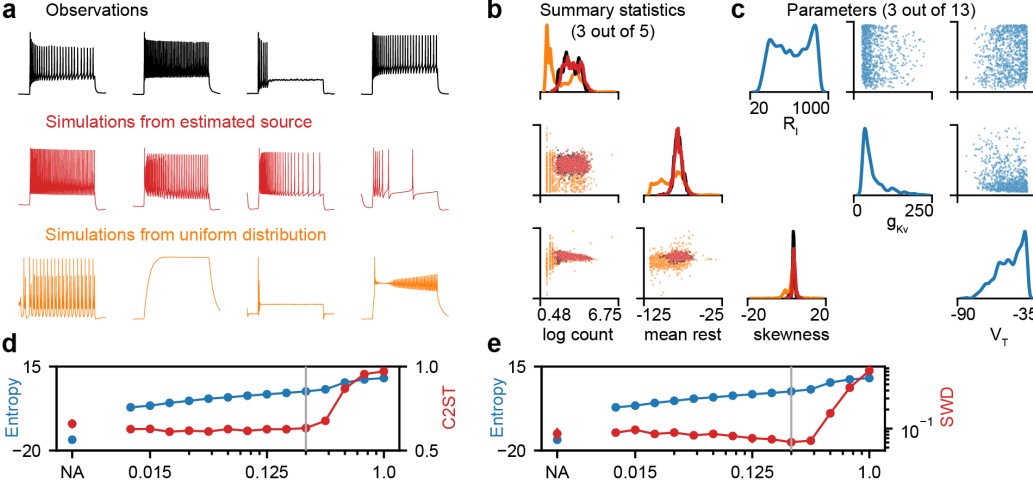

Figure 5: **Source estimation for the single-compartment Hodgkin-Huxley model.** **(a)** Example voltage traces of the real observations of the motor cortex dataset, simulations from the estimated source ($\lambda = 0.25$), and samples from the uniform distribution used to train the surrogate. **(b)** 1D and 2D marginals for three of the five summary statistics used to perform source estimation. **(c)** 1D and 2D marginal distributions of the estimated source for three of the 13 simulator parameters. **(d)** and **(e)** C2ST accuracy and Sliced-Wasserstein distance (lower is better) as well as entropy of estimated sources (higher is better) for different choices of $\lambda$ including $\lambda = 0.25$ (gray line) and without entropy regularization (NA). Mean and standard deviation over five runs are shown.

## 4 Related Work

**Neural Empirical Bayes** High-dimensional source distributions have been estimated through variational approximations to the empirical Bayes problem. Louppe et al. [34] train a generative adversarial network (GAN) [20] $q_\psi$ to approximate the source. The use of a discriminator to compute an implicit distance makes this approach purely sample-based as well. In order to find the optimal $\psi^*$ of the true data-generating process, they augment the adversarial loss with a small entropy penalty on the source $q_\psi$. This penalty encourages low entropy, point mass distributions, which is the *opposite* of our approach. Vandegar et al. [58] take an empirical Bayes approach, and use normalizing flows for both the variational approximation of the source and as a surrogate for the likelihood $p(x|\theta)$. This allows for direct regression on the marginal likelihood, as all likelihoods can be computed directly. Finally, the empirical Bayes problem is also known as "unfolding" in the particle physics literature [10], "population inference" in gravitational wave astronomy [55], and "population of models" in electrophysiology [30]. Approaches have been developed to identify the source distribution, including classical approaches that seek to increase the entropy of the learned sources [50].

**Simulation-Based Inference** The use of variational surrogates of the likelihood of a simulator with intractable likelihood is known as *Neural Likelihood Estimation* in the simulation-based inference (SBI) literature [60, 45, 36, 11]. In neural posterior estimation [44, 35, 21], an *amortized* posterior density estimate is learned, which can be applied to evaluate the posterior of a single observation $x_i \in \mathcal{D}$, if a prior distribution $p(\theta)$ is already known. An intuitive but incorrect approach to source distribution estimation would be to take the *average posterior* distribution over the observations $\mathcal{D}$,

$$G_n(\theta) = \frac{1}{n} \sum_{i=1}^{n} p(\theta|x_i). \tag{7}$$

The average posterior does not always (and typically does not) converge to a source distribution in the infinite data limit, as shown for simple examples in Appendix A.8. Intuitively, the average posterior becomes a worse approximation of a source distribution for simulators that have broader likelihoods. Instead, SBI can be seen as a downstream task of source distribution estimation; once a prior has been learned from the dataset of observations with source estimation, the posterior can be estimated for each new observation individually.

**Generalized Bayesian Inference** Another field related to source estimation is Generalized Bayesian Inference (GBI) [5, 40, 26]. GBI performs distance-based inference, as opposed to targeting the exact Bayesian posterior. Similarly to our work, the distance function used in GBI can be arbitrarily chosen for different tasks. However, GBI is used for single-parameter inference tasks, as opposed to the source distribution estimation task considered in this work. Similarly, Bayesian non-parametric methods [43, 38, 12] learn a posterior directly on the data space which can then be used to sample from a posterior distribution over the parameter space.

## 5   Summary and Discussion

In this work, we introduced Sourcerer as a method to estimate source distributions of simulator parameters given datasets of observations. This is a common problem setting across a range of scientific and engineering disciplines. Our method has several advantages: first, we employ a maximum entropy approach, improving reproducibility of the learned source, as the maximum entropy source distribution is unique while the traditional source distribution estimation problem can be ill-posed. Second, our method allows for sample-based optimization. In contrast to previous likelihood-based approaches, this scales more readily to higher dimensional problems, and can be applied to simulators without a tractable likelihood. We demonstrated the performance of our approach across a diverse suite of tasks, including deterministic and probabilistic simulators, differentiable simulators and surrogate models, low- and high-dimensional observation spaces, and a contemporary scientific task of estimating a source distribution for the single-compartment Hodgkin-Huxley model from a dataset of electrophysiological recordings. Throughout our experiments, we have consistently found that our approach yields higher entropy sources without reducing the fidelity of simulations from the learned source.

**Limitations** In this work, we used the Sliced-Wasserstein distance (and MMD) for the data-consistency term between simulations and observations. In practice, different distance metrics can lead to different estimated sources, depending on its sensitivity to different features. While our method is compatible with any sample-based differentiable distance metric between two distributions, there is still an onus on the practitioner to carefully select a reasonable distance metric for the data at hand. For example, in some cases, it might be appropriate to use a combination of several distance metrics for different modalities of the data. Similarly, there is a dependence on the final regularization strength $\lambda$. Principled methods for defining the regularization strength are desirable, though as we demonstrate, our results are robust to a large range of $\lambda$.

In addition, the method requires a differentiable simulator, which in practice may require the training of a surrogate model, for example, when dealing with a (partially) discrete simulator. While this is a common requirement for simulation-based methods, this could present a challenge for some applications. Finally, in our work, we enforce the maximum entropy principle on the entire (parameter) source distribution. In practice, for example when constructing prior distributions for Bayesian inference, there are other choices, such as the Jeffrey's prior [9].

## Acknowledgements

This work was funded by the German Research Foundation (DFG) under Germany's Excellence Strategy – EXC number 2064/1 – 390727645 and SFB 1233 'Robust Vision' (276693517). This work was co-funded by the German Federal Ministry of Education and Research (BMBF): Tübingen AI Center, FKZ: 01IS18039A and the European Union (ERC, DeepCoMechTome, 101089288). Views and opinions expressed are however those of the author(s) only and do not necessarily reflect those of the European Union or the European Research Council. Neither the European Union nor the granting authority can be held responsible for them. JV is supported by the AI4Med-BW graduate program. JV and GM are members of the International Max Planck Research School for Intelligent Systems (IMPRS-IS). We would like to thank Jonas Beck, Sebastian Bischoff, Michael Deistler, Manuel Glöckler, Jaivardhan Kapoor, Auguste Schulz, and all members of Mackelab for feedback and discussion throughout the project.

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

# A Appendix

## A.1 Software and data

We use PyTorch [46] for the source distribution estimation and hydra [61] to track all configurations. Code to reproduce results is available at https://github.com/mackelab/sourcerer.

## A.2 Simulators and sources

Here we provide a definition of the four benchmark tasks Two Moons (TM), Inverse Kinematics (IK), Simple Likelihood Complex Posterior (SLCP) and Gaussian Mixture (GM), as well as the two high-dimensional simulators, the SIR and Lotka-Volterra model. We also describe the original source distribution used to generate the synthetic observations, and the bounds of the reference uniform distribution on the parameters.

### A.2.1 Two moons simulator

| | |
|---|---|
| **Dimensionality** | $x \in \mathbb{R}^2, \theta \in \mathbb{R}^2$ |
| **Bounded domain** | $[-5, 5]^2$ |
| **Original source** | $\theta \sim \mathcal{U}([-1, 1]^2)$ |
| **Simulator** | $x\|\theta = \begin{bmatrix} r\cos(\alpha) + 0.25 \\ r\sin(\alpha) \end{bmatrix} + \begin{bmatrix} -\|\theta_1 + \theta_2\|/\sqrt{2} \\ (-\theta_1 + \theta_2)/\sqrt{2} \end{bmatrix},$ |
| | where $\alpha \sim U(-\pi/2, \pi/2), r \sim \mathcal{N}(0.1, 0.01^2)$. |
| **References** | Vandegar et al. [58], Lueckmann et al. [37] |

### A.2.2 Inverse Kinematics simulator

| | |
|---|---|
| **Dimensionality** | $x \in \mathbb{R}^2, \theta \in \mathbb{R}^4$ |
| **Bounded domain** | $[-\pi, \pi]^4$ |
| **Original source** | $\theta \sim \mathcal{N}(0, \mathrm{Diag}(\frac{1}{2}, \frac{1}{4}, \frac{1}{4}, \frac{1}{4}))$ |
| **Simulator** | $x_1 = \theta_1 + l_1 \sin(\theta_2 + \epsilon) + l_2 \sin(\theta_2 + \theta_3 + \epsilon) + l_3 \sin(\theta_2 + \theta_3 + \theta_4 + \epsilon),$ |
| | $x_2 = l_1 \cos(\theta_2 + \epsilon) + l_2 \cos(\theta_2 + \theta_3 + \epsilon) + l_3 \cos(\theta_2 + \theta_3 + \theta_4 + \epsilon),$ |
| | where $l_1 = l_2 = 0.5, l_3 = 1.0$ and $\epsilon \sim \mathcal{N}(0, 0.00017^2)$. |
| **References** | Vandegar et al. [58] |

### A.2.3 SLCP simulator

| | |
|---|---|
| **Dimensionality** | $x \in \mathbb{R}^8, \theta \in \mathbb{R}^5$ |
| **Bounded domain** | $[-5, 5]^5$ |
| **Original source** | $\theta \sim \mathcal{U}([-3, 3]^5)$ |
| **Simulator** | $x\|\theta = (x_1, \ldots, x_4), x_i \sim \mathcal{N}(m_\theta, S_\theta),$ |
| | where $m_\theta = \begin{bmatrix} \theta_1 \\ \theta_2 \end{bmatrix}, S_\theta = \begin{bmatrix} s_1^2 & \rho s_1 s_2 \\ \rho s_1 s_2 & s_2^2 \end{bmatrix}, s_1 = \theta_3^2, s_2 = \theta_4^2, \rho = \tanh\theta_5.$ |
| **References** | Vandegar et al. [58], Lueckmann et al. [37] |

### A.2.4 Gaussian mixture simulator

| | |
|---|---|
| **Dimensionality** | $x \in \mathbb{R}^2, \theta \in \mathbb{R}^2$ |
| **Bounded domain** | $[-5, 5]^2$ |
| **Original source** | $\theta \sim \mathcal{U}([0.5, 1]^2)$ |
| **Simulator** | $x\|\theta \sim 0.5\mathcal{N}(x\|\theta, I) + 0.5\mathcal{N}(x\|\theta, 0.01 \cdot I).$ |
| **References** | Sisson et al. [53] |

### A.2.5 SIR model

| | |
|---|---|
| **Dimensionality** | $x \in \mathbb{R}^{50}, \theta \in \mathbb{R}^2$ |
| **Bounded domain** | $[0.001, 3]^2$ |
| **Original source** | $\beta \sim LogNormal(\log(0.4), 0.5) \; \gamma \sim LogNormal(\log(0.125), 0.2)$ |
| **Simulator** | $x\|\theta = (x_1, \ldots, x_{50})$, where $x_i = I_i/N$ equally spaced and $I$ is simulated from $\frac{dS}{dt} = -\beta\frac{SI}{N}, \frac{dI}{dt} = \beta\frac{SI}{N} - \gamma I, \frac{dR}{dt} = \gamma I$ with initial values $S = N - 1, I = 1, R = 0$ and $N = 10^6$. |
| **References** | Lueckmann et al. [37] |

### A.2.6 Lotka-Volterra model

| | |
|---|---|
| **Dimensionality** | $x \in \mathbb{R}^{100}, \theta \in \mathbb{R}^4$ |
| **Bounded domain** | $[0.1, 3]^4$ |
| **Original source** | $\theta' \sim \mathcal{N}(0, 0.5^2)^4$, pushed through $\theta = f(\theta') = \exp(\sigma(\theta'))$, where $\sigma$ is the sigmoid function. |
| **Simulator** | $x\|\theta = (x_1^X, \ldots, x_{50}^X, x_1^Y, \ldots, x_{50}^Y)$, where $x_i^X \sim \mathcal{N}(X, 0.05^2)$, $x_i^Y \sim \mathcal{N}(Y, 0.05^2)$ equally spaced, and $X, Y$ are simulated from $\frac{dX}{dt} = \alpha X - \beta XY, \frac{dY}{dt} = -\gamma Y + \delta XY$ with initial values $X = Y = 1$. |
| **References** | Glöckler et al. [17] |

## A.3 Pseudocode and details on source estimation for benchmark tasks

Pseudocode for Sourcerer is provided in Algorithm 1.

For both the benchmark tasks and high dimensional simulators, sources were estimated from 10000 synthetic observations that were generated by simulating samples from an original previously defined source.

For the benchmark tasks, we used $T = 500$ linear decay steps from $\lambda_{t=0}$ to $\lambda_{t=T} = \lambda$ and optimized the source model using the Adam optimizer with a learning rate of $10^{-4}$ and weight decay of $10^{-5}$. The two high dimensional simulators were optimized with a higher learning rate of $10^{-3}$ and $T = 50$ linear decay steps. In both cases, early stopping was performed when the overall loss in Eq. (4) did not improve over a set number of training iterations.

As a baseline, we compare to Neural Empirical Bayes (NEB) as described in Vandegar et al. [58]. Specifically, we use the biased estimator with 1024 samples per observation ($\mathcal{L}_{1024}$), which are used to compute the Monte Carlo integral. Unlike our Sliced-Wasserstein-based approach, NEB does not operate on the whole dataset of observations directly but attempts to maximize the marginal likelihood per observation and thus uses part of the observations as a validation set. To ensure a fair comparison, we increased the number of observations to 11112 for all NEB experiments, which results in a training dataset of 10000 observations when using 10% as a validation set. For training, we again used the Adam optimizer (learning rate $10^{-4}$, weight decay $10^{-5}$, training batch size 128).

## A.4 Source model

Throughout all our experiments, we use neural samplers as the source models [58]. The sampler architecture is a three-layer multi-layer perceptron with dimension of 100, ReLU activations and batch normalization as our source model. Samples are generated by drawing a sample $s \sim \mathcal{N}(0, I)$ from the standard multivariate Gaussian and then (non-linearly) transforming $s$ with the neural network.

## A.5 Surrogates for the benchmark tasks

We follow Vandegar et al. [58] and train RealNVP flows [13] as surrogates for the four benchmark tasks. For all benchmark tasks, the RealNVP surrogates have a flow length of 8 layers with a hidden dimension of 50.

Surrogates for the benchmark tasks were trained using the Adam optimizer [25] on 15000 samples and simulator evaluations from the uniform distribution over the bounded domain (learning rate $10^{-4}$, weight decay $5 \cdot 10^{-5}$, training batch size 256). In addition, 20% of the data was used for validation.

**Algorithm 1: Sourcerer**

---

**Inputs:** Source model $q_\phi$ constrained on the bounded domain $B_\Theta$, observed dataset $\mathcal{D} = \{x_1, ..., x_n\} \sim p_o(x)$, differentiable model $p(x|\theta)$ to draw samples from (simulator or surrogate), number of samples $m$ to estimate entropy, regularization schedule $\lambda_{t=1}, ..., \lambda_{t=T}$.
**Outputs:** Trained source model $q_\phi(\theta)$.

$t \leftarrow 0$;
**while** *not converged* **do**

$\quad \theta_1, \ldots, \theta_n \sim q_\phi(\theta)$ ;                # sample parameters for pushforward
$\quad x'_i \sim p(x|\theta_i)$ ;                # sample pushforward
$\quad \theta'_1, \ldots, \theta'_m \sim q_\phi(\theta)$ ;                # sample parameters for entropy estimation
$\quad \lambda \leftarrow \lambda_{t=t}$ **if** $t \leq T$ **else** $\lambda_{t=T}$ ;                # schedule lambda
$\quad \mathcal{L} \leftarrow \lambda H(\{\theta'_1, \ldots, \theta'_m\}) + (1 - \lambda)D(\{x_1, \ldots x_n\}, \{x'_1, \ldots, x'_n\})$ ;                # compute loss
$\quad \phi \leftarrow \phi - \text{Adam}(\nabla_\phi \mathcal{L})$ ;                # update source model
$\quad t \leftarrow t + 1$

**return** $q_\phi$

---

To train surrogate models for the SIR and Lotka-Volterra model, we first reduce the simulator dimension in observation space to 25 in both cases. Additionally, we add a small amount of independent Gaussian noise ($\mathcal{N}(X, 0.01^2)$) to the output of the SIR simulator to avoid training the normalizing flow surrogate with simulations from a deterministic likelihood. We then use $10^6$ simulations to train and validate (20% validation set) both surrogate models, again using the Adam optimizer (learning rate $5 \cdot 10^{-4}$, weight decay $5 \cdot 10^{-5}$, training batch size 256).

### A.6   Kozachenko-Leonenko entropy estimator

Our use of neural samplers requires us to use a sample-based estimate of (differential) entropy, since no tractable likelihood is available (see Sec. 2.5).

We use the Kozachenko-Leonenko estimator [28, 3] for a set of samples $\{\theta_i\}_{i=1}^n$ from a distribution $p(\theta) \in P(\Theta)$, given by

$$H(q_\phi) \approx \frac{d}{m} \left[ \sum_{i=1}^n \log(d_i) \right] - g(k) + g(n) + \log(V_d), \tag{8}$$

where $d_i$ is the distance of $\theta_i$ from its $k$-th nearest neighbor in $\{\theta_j\}_{j \neq i}$, $d$ is the dimensionality of $\Theta$, $m$ is the number of non-zero values of $d_i$, $g$ is the digamma function, and $V_d$ is the volume of the unit ball using the same distance measure as used to compute the distances $d_i$.

The Kozachenko-Leonenko estimator is differentiable and can be used for gradient-based optimization. The all-pairs nearest neighbor problem can be efficiently solved in $\mathcal{O}(n \log n)$ [57]. In practice, we find all nearest neighbors by computing all pairwise distances on a fixed number of samples. Throughout all experiments, 512 source distribution samples were used to estimate the entropy during training.

### A.7   Uniqueness of maximum entropy source distribution

Here, we prove the uniqueness of the maximum entropy source distribution (Proposition 2.1). First, however, we demonstrate for a simple example that the source distribution without the maximum entropy condition is not unique.

**Example of non-uniqueness**   Consider the (deterministic) simulator $x = f(\theta) = |\theta|$. Further assume that our observed distribution is the uniform distribution $p(x) = \mathcal{U}(x; a, b)$, where $0 < a < b$. Due the symmetry of $f$, the source distribution $p(\theta)$ for the observed distribution $p(x)$ is not unique. Any convex combination of form $\alpha u_1(\theta) + (1 - \alpha)u_2$, where $u_1(\theta) = \mathcal{U}(\theta; -b, -a)$ and $u_2(\theta) = \mathcal{U}(\theta; a, b)$ and $\alpha \in [0, 1]$ provides a source distribution. The maximum entropy source distribution is unique and is attained if both distributions are weighted equally with $\alpha = 0.5$.

**Proof of Proposition 2.1**    First, let us state Proposition 2.1 in full:

*Let $\Theta \subset \mathbb{R}^{d_\Theta}$ and $\mathcal{X} \subset \mathbb{R}^{d_{\mathcal{X}}}$ be the parameter and observation spaces, respectively. Suppose that $\Theta$ is compact. Let $\mathcal{P}(\Theta) \subset L^1(\Theta)$ and $\mathcal{P}(\mathcal{X}) \subset L^1(\mathcal{X})$ be the set of probability measures on $\Theta$ and $\mathcal{X}$ respectively. Let $Q = \{q | q^\# = p_o \text{ almost everywhere }\} \subset \mathcal{P}(\Theta)$ be the set of source distributions for a given likelihood $p(x|\theta)$ and data distribution $p_o \in \mathcal{P}(\mathcal{X})$. Suppose that $Q$ is non-empty and compact (in the $L^1$ norm topology). Then $q^* = \arg\max_{q \in Q} H(q)$ exists and is unique.*

First, by the compactness assumption on $\Theta$, the (differential) entropy of all $q \in P(\Theta)$ is bounded above (by the entropy of the uniform distribution on $\Theta$), and so in particular it is finite. By the compactness assumption on $Q$, the entropy achieves its supremum of $Q$, that is, there exists a $q^*$ such that $H(q^*) = \arg\max_{q \in Q} H(q)$. To show that $q^*$ is unique (up to $L^1$-null sets), it is sufficient to show two results: (1) that the set $Q$ is a convex set, and (2) that entropy is strictly concave. In this case, if we have two distinct suprema $q_1^*$ and $q_2^*$, then any convex combination of $q_1^*, q_2^*$ is a valid source distribution with higher entropy, causing a contradiction. For the remainder of this proof, we let $q_1$ and $q_2$ be two distinct source distributions. Their convex combination $q = \alpha q_1 + (1 - \alpha) q_2$, $\alpha \in [0, 1]$ is a valid probability distribution supported on both of the supports of $q_1$ and $q_2$.

(1) *Sources distributions are closed under convex combination*: $q$ is also a source distribution, since

$$
\begin{aligned}
q^\#(x) &= \int p(x|\theta) \cdot (\alpha q_1(\theta) + (1-\alpha)q_2(\theta))d\theta \\
&= \alpha \int p(x|\theta)q_1(\theta)d\theta + (1-\alpha)\int p(x|\theta)q_2(\theta)d\theta \\
&= \alpha p_o(x) + (1-\alpha)p_o(x) = p_o(x).
\end{aligned}
\tag{9}
$$

(2) *Entropy is (strictly) concave*: the entropy of $q$ satisfies

$$
\begin{aligned}
H(q) &= -\int (\alpha q_1(\theta) + (1-\alpha)q_2(\theta)) \cdot \log(\alpha q_1(\theta) + (1-\alpha)q_2(\theta))d\theta \\
&\geq -\int [\alpha q_1(\theta)\log(q_1(\theta)) + (1-\alpha)q_2(\theta)\log(q_2(\theta))]d\theta \\
&= \alpha H(q_1) + (1-\alpha)H(q_2),
\end{aligned}
\tag{10}
$$

where we used the fact that the function $f(x) = x \log x$ is convex on $[0, \infty)$, and hence $-f$ is concave. Furthermore, $f(x)$ is strictly convex on $[0, \infty)$, so for any $\theta \in \Theta$, the equality of the integrands

$$
\alpha q_1(\theta) + (1-\alpha)q_2(\theta))\log(\alpha q_1(\theta) + (1-\alpha)q_2(\theta)) = \alpha q_1(\theta)\log(q_1(\theta) + (1-\alpha)q_2(\theta)\log(q_2(\theta)
\tag{11}
$$

holds if and only if $\alpha \in \{0, 1\}$ or $q_1(\theta) = q_2(\theta)$. Since $q_1$ and $q_2$ are assumed distinct, that is, it holds $q_1(\theta) \neq q_2(\theta)$ on a positive measure set, the integral equality in Eq. (10) only holds if $\alpha \in \{0, 1\}$, and thus entropy is strictly concave, which concludes our proof.

$\square$

**Regularized regression as an approximation to constrained optimization**    In practice, we approximate the optimization problem in Eq. (2) with the regularized regression objective in Eq. (3). As a result, we cannot use the result of Proposition 2.1 to guarantee the uniqueness of our solution. However, the dynamic schedule approach to $\lambda$ we use in our work (see Appendix A.3) is similar to the penalty method of approximating solutions to constrained optimization tasks [16, 8]. Future work could use this connection to apply theoretical knowledge of constrained optimization in the source distribution estimation setting.

### A.8    Examples related to the average posterior distribution

In general, the average posterior distribution is not a source distribution. The average posterior distribution is defined in Eq. (7). The infinite data limit is given by $G_n(\theta) \xrightarrow{n \to \infty} G(\theta) = \int p(\theta|x)p_o(x)dx$.

Here, we provide two examples, one based on coin flips, and one based on a Gaussian bimodal likelihood to illustrate this point.

**Coin-flip example**    Consider the classical coin flip example, where the probability of heads (H) follows a Bernoulli distribution with parameter $\theta$. The source distribution estimation problem for this setting would consist of the outcomes of flipping $n$ distinct coins, with potentially different values $\theta_i$.

**Proposition A.1.** *Suppose we have a Beta prior distribution on the Bernoulli parameter $\theta \sim Beta(\alpha, \beta)$ with parameters $\alpha = \beta = 1$, and that the empirical measurements consist of $70\%$ heads, i.e.:*

$$p_o(x) = \begin{cases} 0.7 & x = H \\ 0.3 & x = T \end{cases}$$

*Then the average posterior $G(\theta) = \int p(\theta|x)p_o(x)dx$ is* not *a source distribution for $p_o(x)$.*

*Proof:* Since the Beta distribution is the conjugate prior for the Bernoulli likelihood, the single-observation posteriors are known to be $p(\theta|x = \mathrm{H}) = Beta(2, 1)$ and $p(\theta|x = \mathrm{T}) = Beta(1, 2)$. Hence, the average posterior is

$$G(\theta) = 0.3 \cdot Beta(1, 2) + 0.7 \cdot Beta(2, 1). \tag{12}$$

However, the ratio of heads observed when pushing this distribution through the Bernoulli simulator is

$$
\begin{aligned}
G^{\#}(x = \mathrm{H}) &= \int_0^1 \theta[0.3 \cdot Beta(\theta; 1, 2) + 0.7 \cdot Beta(\theta; 2, 1)]d\theta \\
&= \int_0^1 \theta \left[0.3 \frac{1 - \theta}{B(1, 2)} + 0.7 \frac{\theta}{B(2, 1)}\right] d\theta \\
&= 2\int_0^1 [0.3\theta(1 - \theta) + 0.7\theta^2]d\theta \\
&= 0.3\theta^2 + \frac{2}{3}0.4\theta^3 \Big|_0^1 \approx 0.567 \neq 0.7,
\end{aligned} \tag{13}
$$

where we have used the fact that the Beta function takes the values $B(1, 2) = B(2, 1) = 1/2$. Therefore, the pushforward of the average posterior distribution does not recover the correct ratio of heads, and so it is not a source distribution.

**Gaussian bimodal example**    As another illustrative example to show the differences between average posterior and estimated source, we consider a one-dimensional, bimodal Gaussian likelihood given by $x|\theta \sim 0.5\mathcal{N}(x|\theta - 1, 0.3^2) + 0.5\mathcal{N}(x|\theta + 1, 0.3^2)$ and the source $\mathcal{N}(\theta|0, 0.25^2)$. We use the sbi package [54] and perform neural posterior estimation with the uniform prior $\theta \sim \mathcal{U}([-5, 5])$ to obtain the average posterior and compare it to the source estimated with our approach.

While the estimated source matches the original source closely, the average posterior is visibly different and substantially broader (Fig. A1). As expected, this difference persists when sampling from the average posterior and estimated source to simulate from the likelihood. The pushforward distributions in data space of the original and estimated source match, while the one of the average posterior is again substantially different (Fig. A1).

Additional average posteriors (in comparison to original and estimated source distributions) for the Two Moons and Gaussian mixture are shown in Fig. A6.

### A.9    Details on source estimation for the single-compartment Hodgkin-Huxley model

We use the simulators as described in Bernaerts et al. [2] for our source estimation. This work provides a uniform prior over a specified box domain, which we use as the reference distribution for source estimation. Since the simulator parameters live on different orders of magnitude, we transform the original $m$-dimensional box domain to the $[-1, 1]^m$ cube. Note that this transformation does not affect the maximum entropy source distribution. This is because this scaling results in a constant term added to the (differential) entropy. More specifically, for a random variable $X$ (associated with

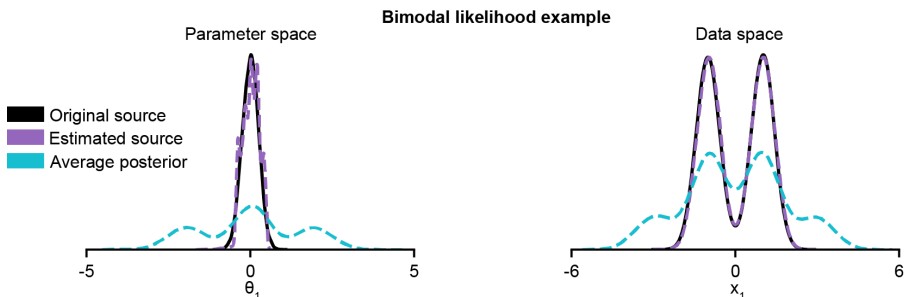

Figure A1: Failure of the average posterior as a source distribution for the bimodal likelihood example. Each of the individual posteriors is bimodal, resulting in an average posterior with 3 modes (left), the secondary modes produce observations which are not observed in the data distribution when pushed through the likelihood (right), and should not be part of the source distribution.

its probability density $p(x)$), the (differential) entropy of $X$ scaled by a (diagonal) scaling matrix $D$ and shifted by a vector $c$ is given by

$$H(DX + c) = H(X) + \log(\det D). \tag{14}$$

The surrogate is trained on $10^6$ parameter-simulation pairs produced by sampling parameters from the uniform distribution and simulating with the sampled parameters. We do not use the simulated traces directly, but instead compute 5 commonly used summary statistics [2, 18]. These are the number of spikes $k$ transformed by a $\log(k + 3)$ transformation (ensuring it is defined in the case of $k = 0$), the mean of the resting potential, and the first three moments (mean, variance, and skewness) of the voltage during the stimulation.

As our surrogate, we choose a deterministic multi-layer perceptron, because we found that the internal noise has almost no noticeable effect on the summary statistics, so that the likelihood $p(x|\theta)$ is essentially a point function. We are able to make this choice because the sample based nature of our source distribution estimation approach is less sensitive to sharp likelihood functions, whereas likelihood-based approaches could struggle with such problems.

The multi-layer perceptron (MLP) surrogate has 3 layers with a hidden dimension of 256. ReLU activations and batch normalization were used. Training of the MLP was done with Adam (learning rate $5 \cdot 10^{-4}$, weight decay $10^{-5}$, training batch size 4096). Again, 20% of the data were used for validation.

## A.10   Computational Resources

All numerical experiments reported in this work were performed on GPU using an NVIDIA A100 GPU. A single source estimation run for a benchmark task using the Sourcerer approach (for one value of $\lambda$) took approx. 30 seconds. In comparison, learning the source using NEB for the same task took approx. 2 minutes (see Table A1). A source estimation run for Sourcerer on the high-dimensional tasks took approx. 10 min. When the observations are high-dimensional, training a surrogate (if required) makes up the majority of the computational cost. For the Hodgkin-Huxley task, training a surrogate took approx. 20 minutes, after which estimating the source distribution with Sourcerer took approx. 30 seconds.

Table A1: **Wall-clock runtime comparison between Sourcerer and NEB.** Time in seconds measured on an Nvidia A100 GPU. Average and standard deviation are shown over 5 runs. For all three settings (Sourcerer with and without entropy regularization, NEB), surrogate models for the benchmark simulators were used. Sourcerer converges noticeably faster than the NEB baseline.

| Method | Sur. (w/o reg.) | Sur. (with reg.) | NEB |
|--------|-----------------|------------------|-----|
| TM | 29.4 (8.5) | 63.9 (10.1) | 145.2 (13.9) |
| IK | 28.5 (6.9) | 66.7 (10.0) | 116.8 (22.6) |
| SLCP | 71.7 (12.8) | 53.1 (12.2) | 91.6 (9.9) |
| GM | 26.6 (5.4) | 46.2 (9.2) | 98.5 (15.5) |

## A.11 Supplementary figures

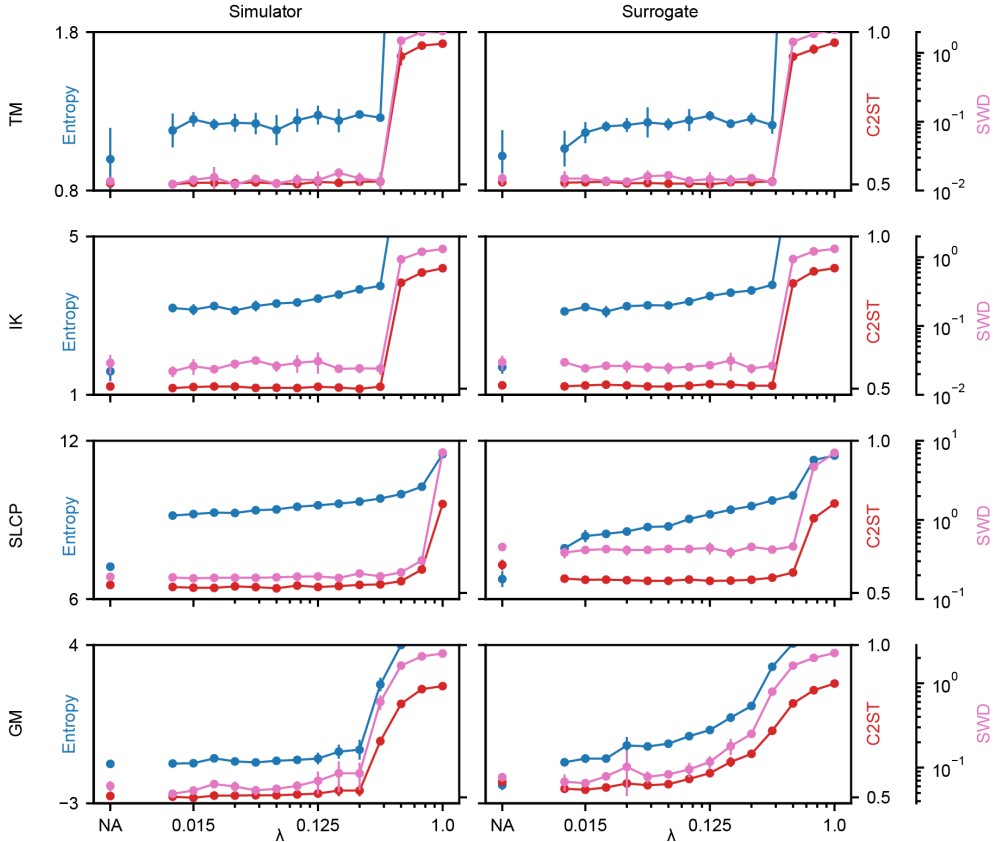

Figure A2: Extended results for source distribution estimation on the benchmark tasks (Fig. 3) for different choices of $\lambda$. In addition to the C2ST accuracy and entropy, here the Sliced-Wasserstein distance (SWD) between the observations and the pushforward distribution of the estimated source is shown. Mean and standard deviation were computed over five runs.

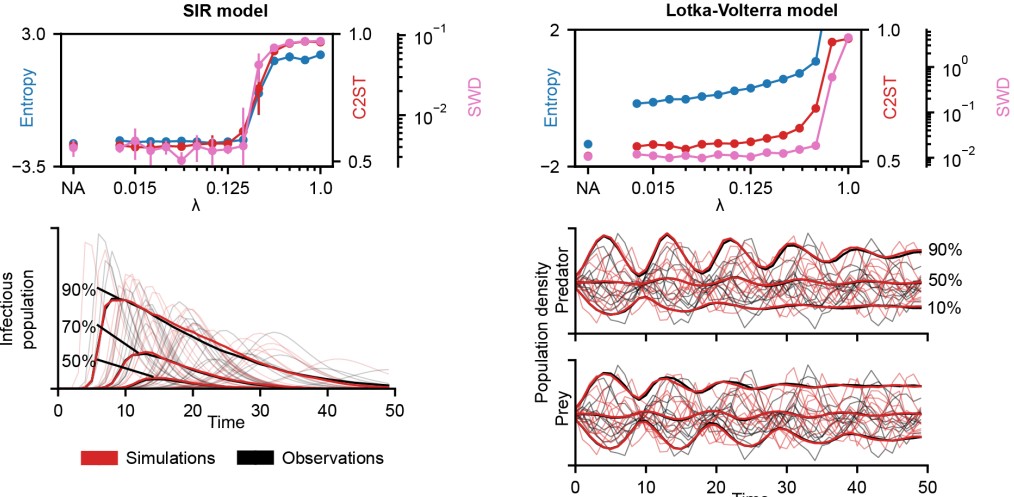

Figure A3: Extended results for source distribution estimation on the differentiable SIR and Lotka-Volterra models (Fig. 4). In addition to the Sliced-Wasserstein distance (SWD), the C2ST accuracy between the observations and the pushforward distribution of the the estimated source is shown. Despite the high-dimensional data space of the simulators (50 and 100 dimensions), the estimated sources achieve a good C2ST accuracy (below 60%) for various choices of $\lambda$. Mean and standard deviation were computed over five runs. Additionally, percentile values of all samples computed per time point between simulations (simulated with parameters from the estimated source) and observations (simulated with parameters from the original source) closely match.

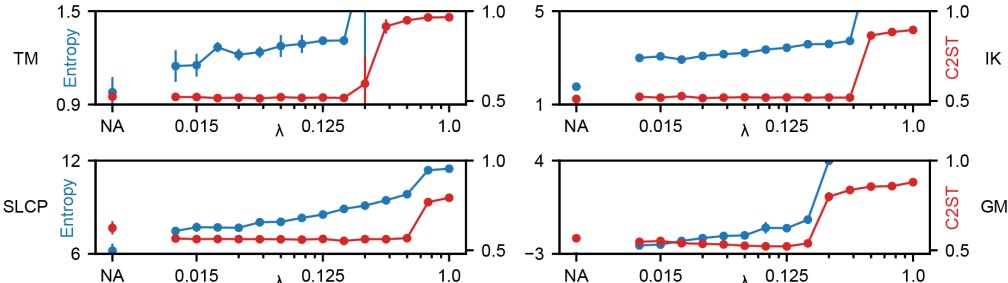

Figure A4: Sourcerer with Maximum Mean Discrepancy (MMD) as the differentiable, sample-based distance. We use MMD with an RBF kernel and the median distance heuristic for selecting the kernel length scale. Source estimation is performed without (NA) and with entropy regularization for different choices of $\lambda$. For these tasks, MMD produces similar results to the previously used SWD (Fig. 3b). These results show that Sourcerer is compatible with other sample-based, differentiable distances other than the SWD. For all cases, mean C2ST accuracy between observations and simulations (lower is better) as well as the mean entropy of estimated sources (higher is better) over five runs are shown together with the standard deviation.

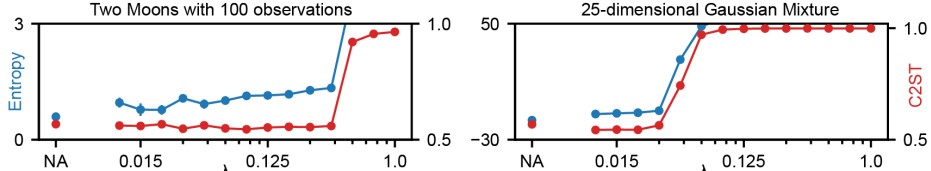

Figure A5: Experiments with less observations and higher-dimensional sources. Source estimation without (NA) and with entropy regularization for different choices of $\lambda$. For the Two Moons task, the number of observations was reduced from 10000 to 100. For the Gaussian Mixture task, the dimensionality was increased from 2 to 25. These results show that Sourcerer is robust to small datasets of observations, and can estimate high-dimensional source distributions. For all cases, mean C2ST accuracy between observations and simulations (lower is better) as well as the mean entropy of estimated sources (higher is better) over five runs are shown together with the standard deviation.

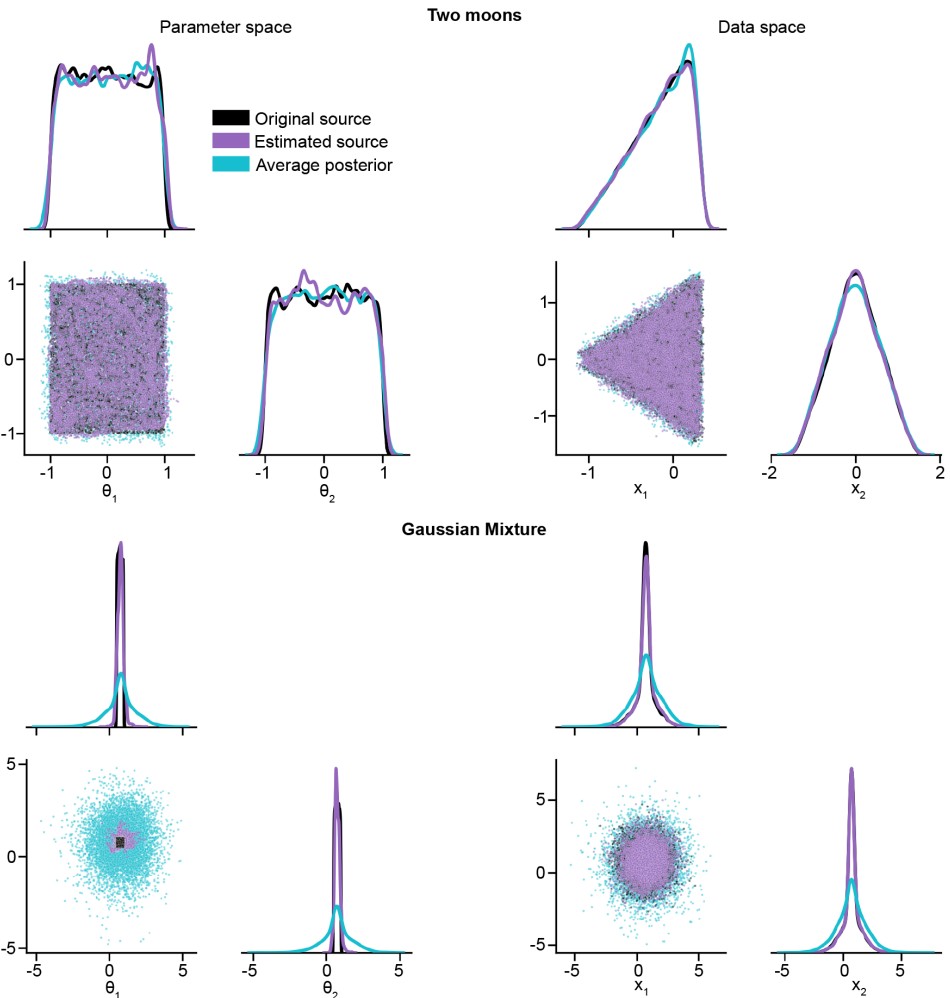

Figure A6: Original and estimated sources distributions as well as average posterior distribution for Two Moons and Gaussian Mixture simulator with uniform prior $\theta \sim \mathcal{U}([-5,5]^2)$. For simulators for which the likelihood is unimodal and narrow, such as the Two Moons simulator, the average posterior can be a good approximation of a source distribution. However, for simulators where the likelihood is broader, such as the Gaussian Mixture simulator, the average posterior is too broad, and does not reproduce the data distribution $p_o$ well, when compared to estimates of source distributions.

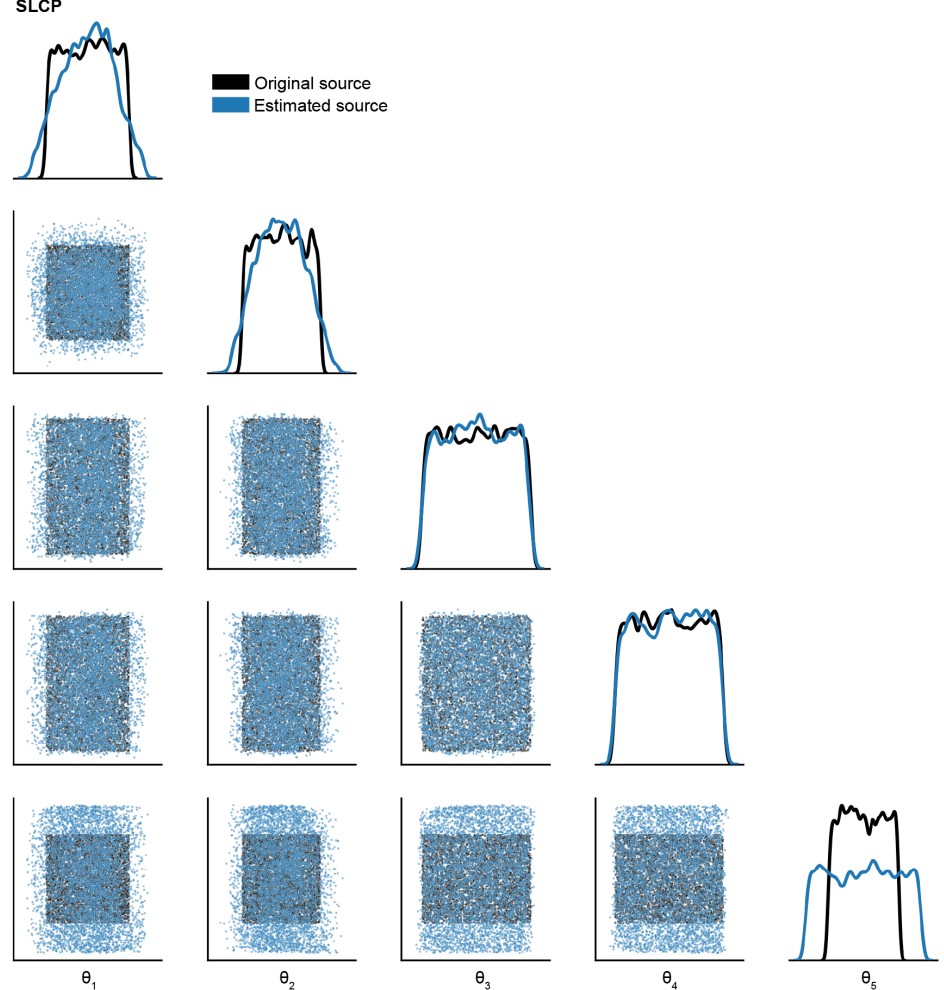

Figure A7: Original and estimated source distributions for the benchmark SLCP simulator. The estimated source has higher entropy than the original source.

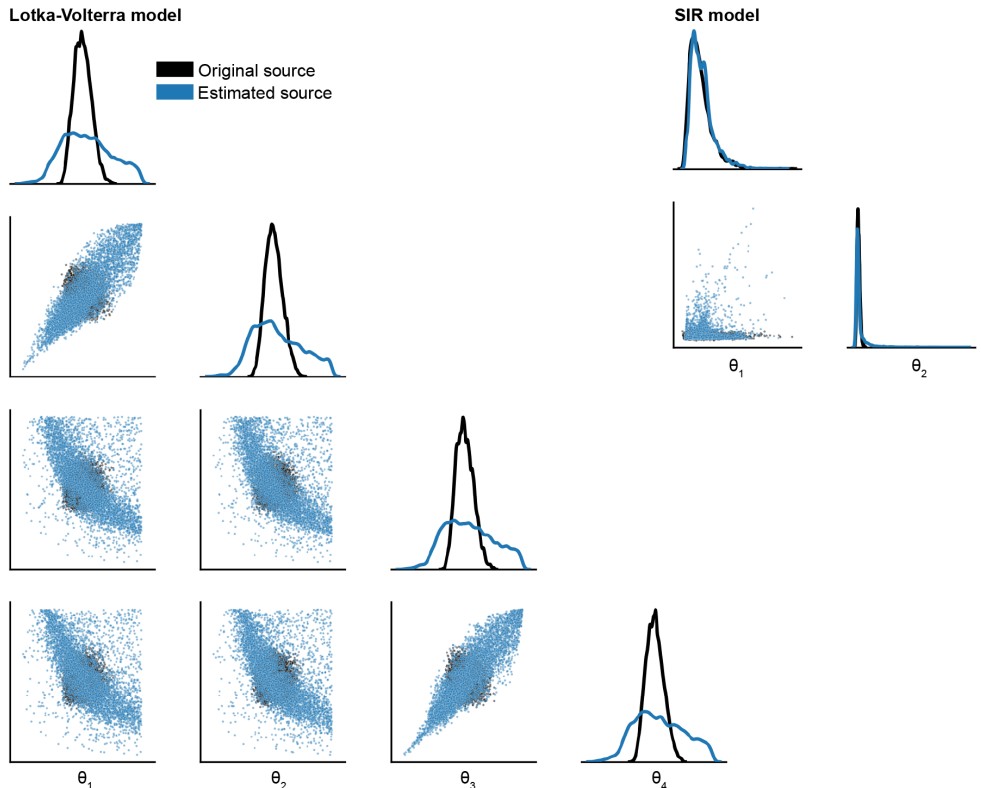

Figure A8: Original and estimated source distributions for the SIR and Lotka-Volterra model. For the Lotka-Volterra model, the estimated source has higher entropy than the original source.

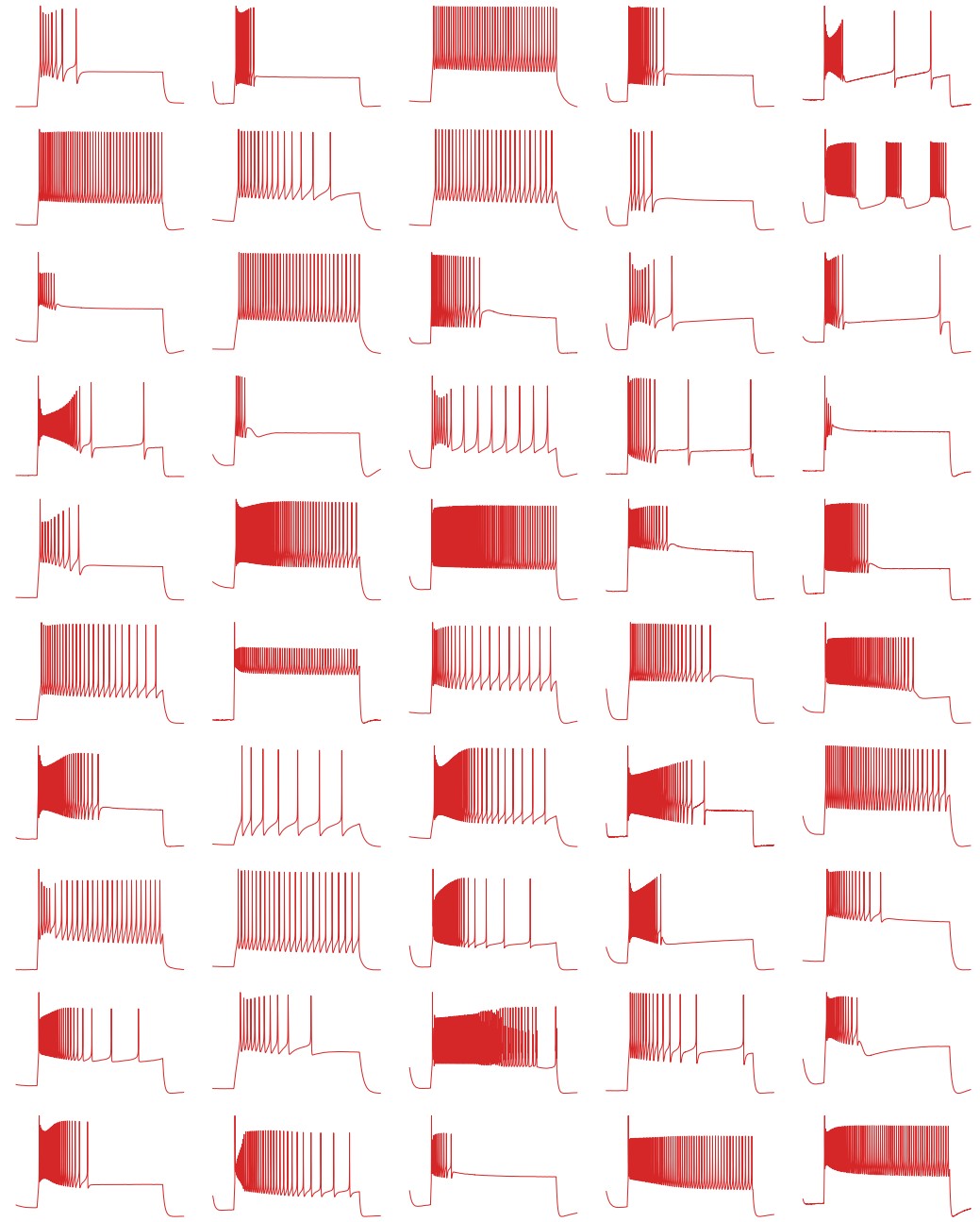

Figure A9: 50 random example traces produced by sampling from the estimated source and simulating with the Hodgkin-Huxley model.

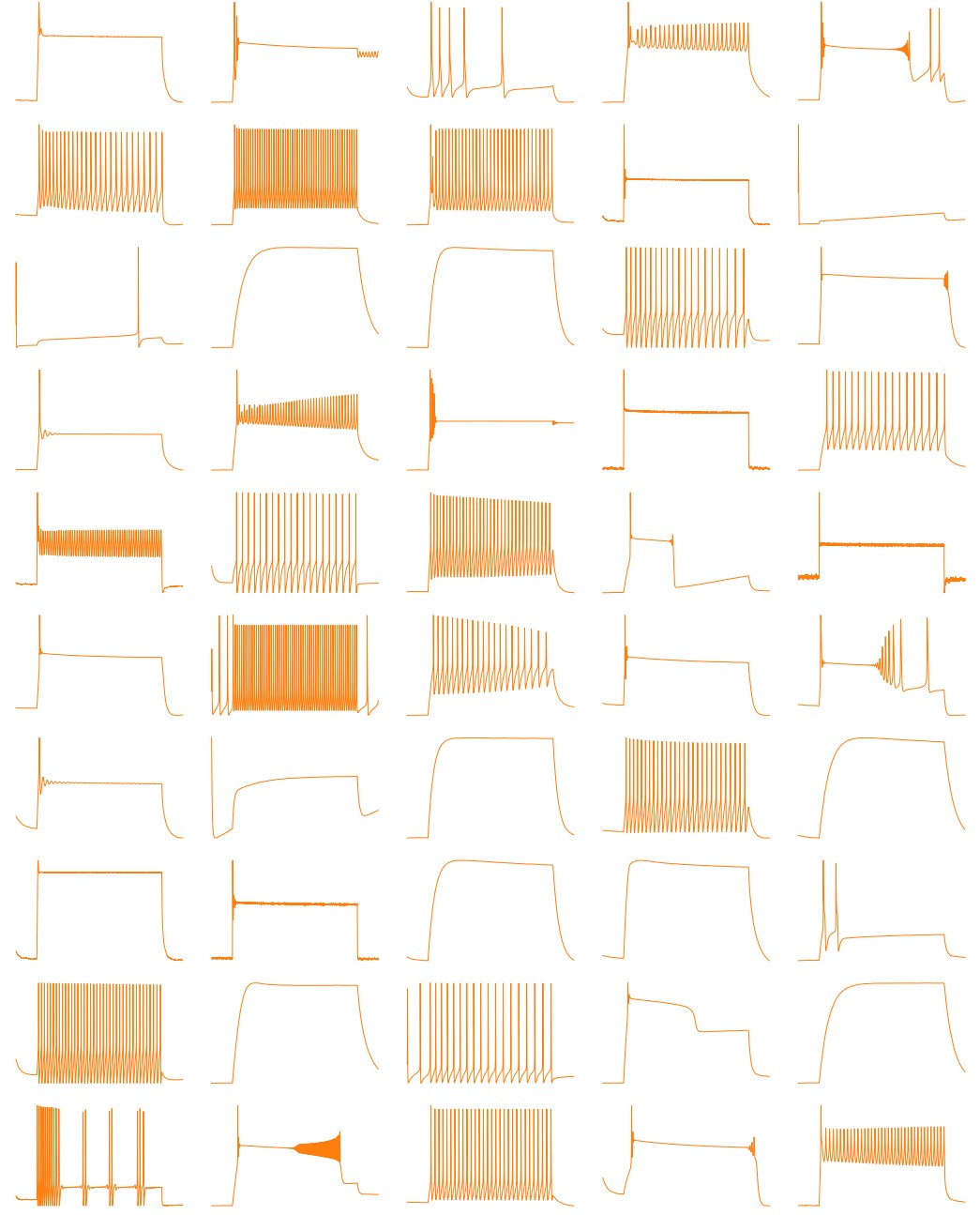

Figure A10: 50 random example traces produced by sampling from the uniform distribution over the box domain and simulating with the Hodgkin-Huxley model.

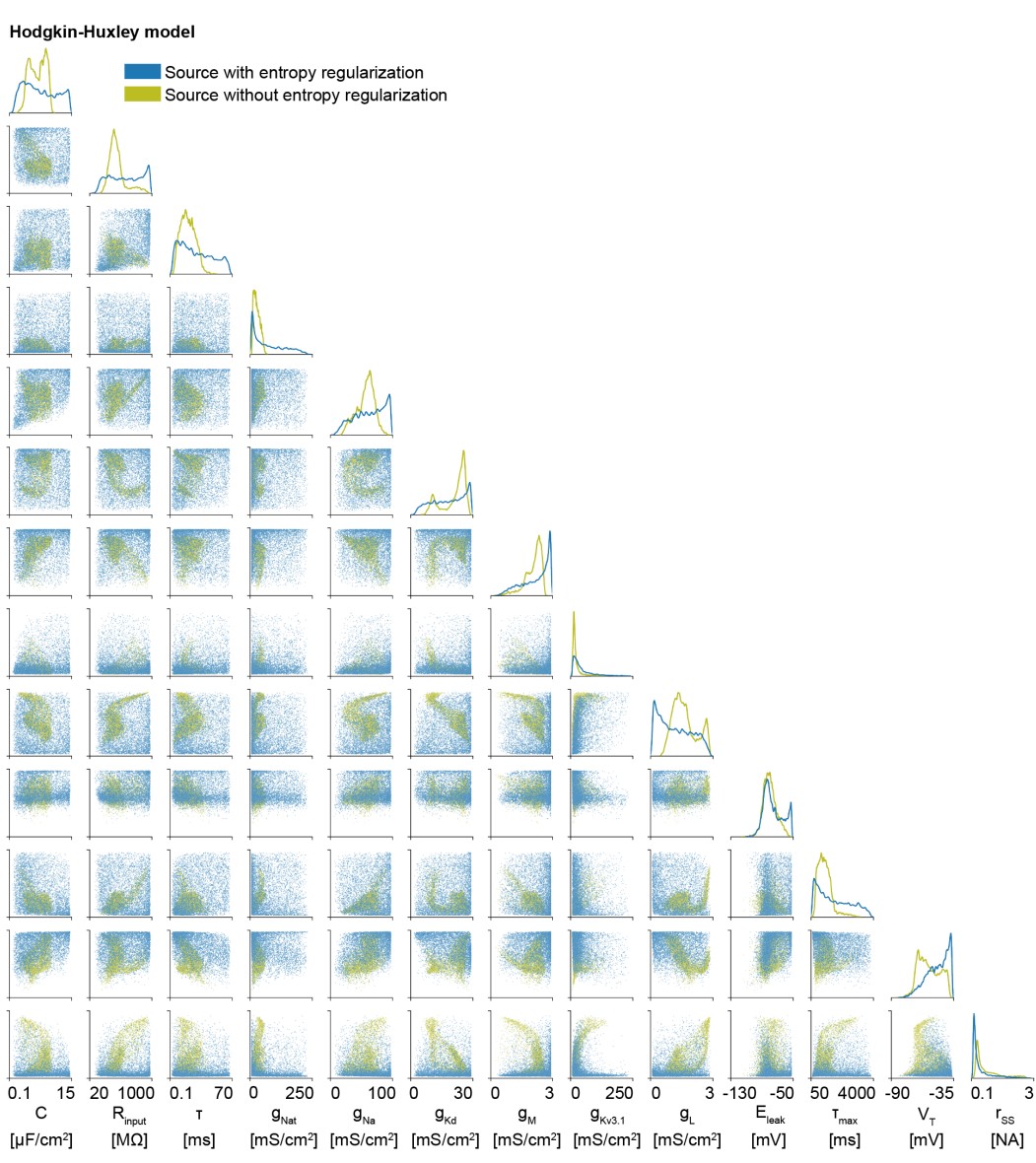

Figure A11: Estimated sources using for Hodgkin-Huxley task with the entropy regularization ($\lambda = 0.25$) and without the entropy regularization. Without, many viable parameter settings are missed, which would have significant downstream effects if the learned source distribution is used as a prior distribution for inference tasks.

