# OpenReview forum: "Sourcerer: Sample-based Maximum Entropy Source Distribution Estimation"
_NeurIPS.cc/2024/Conference — NeurIPS 2024 poster_

### Official Review · Reviewer_8Zuo · 2024-06-19

**Soundness:** 3
**Presentation:** 4
**Contribution:** 3
**Rating:** 7
**Confidence:** 3

**Summary:**

This paper introduces a new algorithm for source estimation, the task of estimating a model's parameters probability distributions consistent with observations. In opposition to previous algorithms, the proposed method is sampling-based and can hence be used when the model is a simulator, implicitly defining the likelihood even if the observations are high-dimensional. The authors highlight that several probability distributions might be consistent with the observations and advocate favoring the one with the highest entropy. Their method provides a way to target high-entropy solutions and shows that this new objective leads to a unique optimum.

**Strengths:**

### Originality
* The method is new to me.

### Quality
* I did not identify flaws in the mathematical derivations.
* The method is evaluated on a wide variety of benchmarks, and several runs have been made. The experiments showcase that the entropy regularization indeed leads to higher entropy sources.

### Clarity
* All the necessary details are provided
* The paper is well articulated.
* The method is clearly explained
* It is clearly stated how the method fits in the literature.
* Figures are clean.

### Significance
* Source estimation is a task that has received attention in recent years.
* This paper fills a gap in the literature by providing a method that works with high-dimensional simulators.

**Weaknesses:**

### Originality
* I have no concerns regarding originality.

### Quality
* I find that the Lotka-Volterra and SIR experiments are not totally convincing. First, it lacks comparison with NEB, which is claimed to perform poorly in such settings. I agree that NEB will probably provide poor results, but this should still be shown empirically, in my opinion. Second, the Wasserstein distance is hard to interpret. It is claimed that a low value is observed, but that is not clear to me why this value can be considered a low value as there are no baselines to compare with.

### Clarity
* I have no concerns regarding clarity.

### Significance
* I have no concerns regarding significance.

**Questions:**

* Is there something that prevents you from comparing against NEB in Figure 4?

**Limitations:**

The authors have adequately addressed the limitations.

---

> ### Author Rebuttal · Authors · 2024-08-06
>
> We thank the reviewer for the positive feedback of our work.
>
> **W/Q1: "Is there something that prevents you from comparing against NEB in Figure 4?"** We thank the reviewer for their suggestion, and have now compared Neural Empirical Bayes (NEB) with Sourcerer on the high-dimensional simulators (SIR and Lotka-Volterra model). Results for this experiment can be found in the provided rebuttal supplement (Fig. R2). In summary, we find that Sourcerer performs better on both tasks: on SIR, Sourcerer achieves a C2ST accuracy (lower is better) of 55% between data and predictive simulations of learned source distribution, compared to 76% for NEB; similarly, on Lotka-Volterra, Sourcerer achieves a C2ST of 56%, compared to 62% for NEB. Thus, Sourcerer is demonstrably better than NEB on these high-dimensional tasks.
>
> To perform this experiment, we trained surrogates for each simulator. While this was not necessary for Sourcerer, attempting to perform NEB on the differentiable likelihood of the differential equation simulators was computationally infeasible due to the large minibatches required for NEB. To enable surrogate training, we reduced the observation dimensionality of both models to 25 (instead of 50 and 100 respectively), and added Gaussian noise to the previously deterministic SIR simulator. For a fair comparison, we use the same surrogate model for both Sourcerer and NEB.
>
> We argue that Sourcerer's improved performance over NEB on these tasks is due to the different ways in which the two approaches use the surrogate model - NEB is sensitive to the exact likelihood, and thus may suffer from a lack of robustness to misspecification as in the case of a surrogate model. Our method, on the other hand, requires only samples, and thus the exact likelihoods produced by the surrogate model do not have a detrimental effect.
>
> **W1.1 "The Wasserstein distance is hard to interpret. It is claimed that a low value is observed, but that is not clear to me why this value can be considered a low value as there are no baselines to compare with"**: We agree with the reviewer that the numerical value of the Sliced-Wasserstein Distance (SWD) is hard to interpret in isolation. We had therefore computed the expected SWD between two sets of independently generated observations (simulated from the true source distribution). This minimum achievable distance is indicated by the black dotted lines in Figure 4 of our original submission. We found that for both models, the distance between simulations (from the estimated source) and observations is very close to the expected distance between simulations, and will emphasize this in the revision. Finally, we also provide the more interpretable C2ST accuracies for the SIR and Lotka-Volterra experiments in the Appendix (Fig. 8). In both cases, Sourcerer achieves a C2ST accuracy close to 50%.

---

> > ### Comment · Reviewer_8Zuo · 2024-08-12
> >
> > Thanks for the update.
> >
> > I have read the rebuttal and the other reviews and keep my score unchanged.

---

> > > ### Author Response · Authors · 2024-08-14
> > >
> > > We thank the reviewer for their response.

---

### Official Review · Reviewer_JA3k · 2024-07-05

**Soundness:** 3
**Presentation:** 3
**Contribution:** 3
**Rating:** 6
**Confidence:** 3

**Summary:**

This paper deals with the problem of identifying the distribution of a source variable $s$ that generates observations $x$.

## The problem

The authors propose to minimize a classical objective consisting of two terms:

1. A reconstruction term, that encourages the recovered source distribution to induce a distribution over observations that is close to the data distribution, in a certain divergence (here, the sliced Wasserstein)

2. A regularization term, that encourages the recovered source distribution to be close to a reference distribution, in a certain divergence (here, the reverse KL).

Many different source distributions can minimize the first term alone, which motivates the second term to contrain the solution space.

## The probability model

The probability model $p(x)$ can be expressed in terms of two other model distributions: the source distribution $p(s)$ and the likelihood $p(x | s)$.

Term (1) uses a (sliced) Wasserstein divergence. Computing this divergence requires data samples and samples from the model in two steps:  $s \sim p(s)$ then $x | s \sim p(x | s)$. Differentiating through this divergence requires the probability model, and therefore $p(s)$ and $p(x | s)$, to be samplable and the sampling process should be differentiable.

Term (2) uses a reverse KL divergence. Computing this divergence requires evaluating the densities of the model and reference distributions, as well as samples from the model in two steps:  $s \sim p(s)$ then $x | s \sim p(x | s)$. Differentiating through this divergence requires the probability model, and therefore $p(s)$ and $p(x | s)$, to be samplable and the sampling process should be differentiable. Additionally, the model density should be known or else a differentiable approximation of the divergence (the authors use the Kozachenko-Leonenko estimator) should be available.

## Numerics

The authors validate their method using synthetic data where the source is known, and real data where the observations are neurophyisiological recordings and the likelihood is given by a mechanistic model.

**Strengths:**

The presentation is clear and well-explained.

**Weaknesses:**

Overall, I have two concerns.

1. The computational constraints on the model are heavy

The choices of divergences (Sliced-Wasserstein and reverse KL) in the cost function impose a number of computational constraints of the model.

Term 1 in the cost uses a vanilla Sliced-Wasserstein divergence estimator that is sample-based. This means the likelihood model $p(x | s)$ should be samplable and the sampling process should be differentiable. The authors propose to train a differentiable approximator of a black-box simulator. To be fair, the authors acknowledge that this is a general issue beyond their paper.

Term 2 in the cost uses a vanilla KL divergence estimator which requires knowing the density of the model, but the authors circumvent this by using another estimator named "Kozachenko-Leonenko". Given that the authors already used a sample-based divergence in Term 1, wouldn't using another sample-based divergence in the Term 2 (e.g. the Sliced-Wasserstein again, or an MMD) between the model and reference distributions not add any further computational constraints on the model?

2. The evaluation procedure or goal remains unclear to me

The framework of minimizing a cost made up of a reconstruction and regularization term is well-studied in optimization. Once could imagine two natural evaluation procedures:
- how well the true sources are recovered? But the authors are clear that designing an identifiable probability model where the sources can be recovered (up to acceptable indeterminacies) is not the goal here.
- how efficient is their method on a new test dataset, as compared to other similar methods?

It seems to be that the authors answer the latter, but it is not clear to me what are the other methods to benchmark against.

**Questions:**

Could the authors address the weaknesses part?

**Limitations:**

Yes.

---

> ### Author Rebuttal · Authors · 2024-08-06
>
> We thank the reviewer for their feedback. We address the reviewer's concerns below.
>
> **W1: "The computational constraints on the model are heavy"**: The reviewer raises concerns about computational constraints on the model, which we clarify here for the two terms in the cost function:
>
> First, the entropy/KL-divergence term (term 2) imposes negligible computational constraints on the model because it is computed on samples in parameter space, and not in data space. In particular, given samples $\theta_i$ from our source model, we only need to compute their nearest-neighbor distances $d_i$ to estimate the entropy (Eq. 8 of our submission). Thus, its computation is completely independent of the simulator or surrogate. Hence, this concern is not a limitation of the method.
> In addition, while other distances (such as Sliced-Wasserstein or MMD, as suggested by the reviewer) can be used to regularize the source in the second term, we emphasize that the choice of entropy (KL-divergence wrt. uniform distribution) stems from the original motivation of the work (i.e., the theoretical justification in Prop 2.1.)
>
> Second, we agree with the reviewer that requiring the simulator to be differentiable for term 1 is a computational constraint on our model. However, we emphasize that our approach is an improvement over prior work (in particular, Vandegar et al. (2020)) in this regard. As the reviewer identified, the requirement to train surrogates for non-differentiable models is a broader limitation of the field. However, typical works in the field require training a surrogate that has a tractable _and_ differentiable likelihood. We only require differentiability. We do not require tractable likelihoods, which reduces the computational constraints. This is a relative strength of our method relative to previous approaches, not a limitation.
>
> [Vandegar et al.] - Neural Empirical Bayes: Source distribution estimation and its applications to simulation-based inference. In International Conference on Artificial Intelligence and Statistics, 2020.
>
> Finally, in terms of computational cost, the choices we make for the two terms in the loss function are highly efficient. For the mismatch term (term 1), the Sliced-Wasserstein distance is linearithmic in the number of observations, and thus scales well to millions of observations. Furthermore, the distance computation of individual 1-D slices is embarrassingly parallelizable. Our use of Kozachenko-Leonenko estimators to estimate entropy (term 2) is also scalable, since computing the estimate requires solving the all-nearest-neighbors problem, which can also be solved in linearithmic time. We now perform a runtime comparison between NEB and Sourcerer for the benchmark tasks, and find that Sourcerer is significantly faster (Table R1).
>
>
> **W2: "The evaluation procedure or goal remains unclear to me"**: We apologize for not making our evaluation procedure sufficiently clear. For the benchmark tasks as well as the SIR and Lotka-Volterra experiment, we always evaluate the match between simulations and observations on a second, unseen ‘test’ set of observations that was generated independently using samples from the ground truth source distribution. Thus, we already do the evaluation in the way the reviewer asks, and just failed to point this out in the manuscript. We will clarify in the revised version.
>
> More generally, our evaluation is always based on two criteria:
> 1. How well do the simulations (generated by the estimated source) match the set of observations? We measure this match using Classifier Two Sample Tests (C2ST) and the SWD.
> 2. How high is the entropy of the estimated source? We measure this by estimating the differential entropy with samples from the source using the Kozachenko-Leonenko estimator.
>
> We benchmark our  method against Neural Empirical Bayes (NEB), a state-of-the-art approach for source distribution estimation. For the benchmark tasks, we showed that Sourcerer obtains sources that are comparable to NEB in terms of simulation fidelity (point 1), while having a higher entropy (point 2). We now also perform an additional comparison on the high-dimensional data tasks of SIR and LV (Fig. 2), and find that our method also significantly outperforms NEB in terms of simulation fidelity (point 1). On SIR, Sourcerer achieves a C2ST accuracy (lower is better) of 55% between data and predictive simulations of learned source distribution, compared to 76% for NEB; similarly, on Lotka-Volterra, Sourcer achieves a C2ST of 56%, compared to 62% for NEB.

---

> > ### Comment · Reviewer_JA3k · 2024-08-11
> > **Answer to authors**
> >
> > I thank the authors for their clarifications and am happy to raise my score accordingly.

---

> > > ### Author Response · Authors · 2024-08-14
> > >
> > > We thank the reviewer for their response.

---

### Official Review · Reviewer_J2pY · 2024-07-10

**Soundness:** 3
**Presentation:** 3
**Contribution:** 2
**Rating:** 5
**Confidence:** 2

**Summary:**

This paper proposes an approach to estimating a maximum-entropy source distribution (akin to a prior distribution over simulator parameters) for a given set of observations and simulation model. Their method assumes a differentiable simulator that may be deterministic or stochastic, and it uses neural samplers to approximate the prior and a variational objective that encourages proximity of both the marginal likelihood to the true data distribution and the estimated source distribution to some known prior distribution (which may be uniform, corresponding to entropy regularisation). In particular, proximity to the true data distribution is measured with a sliced Wasserstein distance, due to its fast computation and differentiability (preserving the differentiable pipeline). The authors present experiments on four benchmark tasks, before extending to higher dimensional and more complex examples such as the single-compartment Hodgkin-Huxley model.

**Strengths:**

*Originality*

The authors consider using a sample based loss to capture mismatch between the simulated and true data distributions, in contrast to using likelihood-based notions of distance. This is useful since simulation models often lack tractable likelihood functions.

*Quality*

The experimental section presents good and extensive empirical testing on a number of toy benchmark models, in addition to two additional more complex simulation models. Their approach is also well-motivated via Proposition 2.1.

*Clarity*

The clarity of writing is generally good. There are a few minor errors in the writing that made it not absolutely clear all the way through:
- The last sentence of the first paragraph of Section 1 isn't a full sentence (or, if it is, then it not well-written because I have read it multiple times and could not turn it into a full sentence in my head).
- There is a subscript $\phi$ missing in Equation 4.

*Significance*

I think this paper will be of some interest to the community and the techniques presented used by practitioners.

**Weaknesses:**

My main concern is that the work presented looks like it is not a large or significant contribution beyond what is already present in the literature. From my perspective, it looks like someone could reasonably argue that the contribution is just to use a different sample-based loss function to capture the discrepancy between the data distribution and simulator's marginal likelihood. It also isn't quite clear to me how exactly the approach that uses the surrogate model (Sur. in Table 1) is different from what is presented in Vandegar et al. (2020).

**Questions:**

Could the authors provide some more detail and clarification on how their approach differs/relates to/builds on previous works please? Ideally at least those papers already cited such as Vandegar et al. (2020), but I think some additional related literature is also missing, such as references [1] (relevant since it discusses variational approaches to targeting generalised Bayesian posteriors) and [2] (relevant to the problem of source estimation in complex simulation models) below. Is there anything that we lose by using something like the Sliced Wasserstein Distance instead of a likelihood-based discrepancy?

[1] _Knoblauch, J., Jewson, J., & Damoulas, T. (2022). An optimization-centric view on Bayes' rule: Reviewing and generalizing variational inference. Journal of Machine Learning Research, 23(132), 1-109._

[2] _Dyer, J., Quera-Bofarull, A., Bishop, N., Farmer, J. D., Calinescu, A., & Wooldridge, M. (2024, May). Population Synthesis as Scenario Generation for Simulation-based Planning under Uncertainty. In Proceedings of the 23rd International Conference on Autonomous Agents and Multiagent Systems (pp. 490-498)._

**Limitations:**

The authors provide a good discussion of the limitations towards the end of the paper. Perhaps something additional to discuss would be limitations associated with simulators that involve discrete randomness, which (as far as I can tell) aren't tested in this paper but which often appears in simulation modelling and presents an additional complication when the requirement is that sampling from the simulator is a differentiable operation (see e.g. [1] below). I also do not see any problems related to the broader social impact of this work.

[1] _Arya, G., Schauer, M., Schäfer, F., & Rackauckas, C. (2022). Automatic differentiation of programs with discrete randomness. Advances in Neural Information Processing Systems, 35, 10435-10447._

---

> ### Author Rebuttal · Authors · 2024-08-06
>
> We thank the reviewer for their feedback and careful reading. We address the main concerns and questions raised by the reviewer below. Thank you for identifying typos, we will correct them.
>
> **W1: "My main concern is that the work presented looks like it is not a large or significant contribution beyond what is already present in the literature"**: We summarize the two main contributions of our work:
>
> First, we provide an approach to resolve the ill-posedness of the source estimation problem by using the maximum entropy principle, and show that it leads to a unique solution.  Empirically, we demonstrate that our approach can estimate high-entropy sources without sacrificing the quality of the match between simulations (from the estimated source) and observations.
> Second, our approach is fully sample-based. This provides significant computational advantages over previous methods, since our method can use differentiable simulators directly, without training surrogate models.
>
> **W1.1: "Detailed comparison to Neural Empirical Bayes"**: Our work makes conceptual, theoretical, and empirical contributions that go beyond Neural Empirical Bayes (NEB, Vandegar et al. (2020)):
>
> First, a limitation of NEB - as directly acknowledged in their publication (Section 2, third paragraph) - is that the source distribution estimation problem is ill-posed. This ill-posedness is not addressed by the NEB method. In our work, we propose to use the maximum entropy principle as a way to resolve this ill-posedness, and make the theoretical contribution of showing that the maximum entropy source distribution is unique (Prop. 2.1).  Furthermore, we demonstrate empirically (Table 1, Figures 3,4,5) that our method finds higher entropy source distributions than NEB, without any decline in the predictive performance of the estimated source distributions.
>
> Second, our approach is entirely sample-based, i.e. it does not require access to likelihoods (or surrogate models with likelihoods).  We show the advantages of our sample-based approach over NEB in the context of high-dimensional, differentiable simulators. In this setting, our method (unlike NEB) does not require the training of surrogate models. We now additionally show (Fig. R2) that our sample-based objective is more robust than NEB even in the case where a surrogate is available for higher-dimensional tasks. For the SIR task, our method achieves a C2ST accuracy (lower is better) of 55% between data and predictive simulations of the learned source distribution, compared to 76% for NEB; similarly, on Lotka-Volterra, our method achieves a C2ST of 56%, compared to 62% for NEB. Thus, our method is demonstrably superior to NEB on these high-dimensional tasks.
>
> **W1.2: "Comparison to additional works"**: The reviewer also asked us to contrast our approach with previous work.
>
> Regarding Knoblauch et al. (2022) [1]:  The idea of using other distance measures between distributions is the key approach of Generalized Bayesian Inference (GBI) methods, as discussed in our related work and in [1]. Our method shares similarities with GBI in that it uses non-likelihood-based objective functions. However, we emphasize that our work is concerned with source distribution estimation. This is a different problem than inference, in that we seek to identify a distribution of parameters that is consistent with a population of independent observations (many to many), rather than to explain a single or multiple observations from a single parameter (i.e., posterior inference, one to one or many to one). These are not technical differences, but rather that these methods simply address different goals.
>
> Regarding Dyer et al. (2024) [2]: This problem setting is different from ours. In this paper, the authors seek to find a distribution that produces simulations “matching closely a target, user-specified scenario”. This is different from the source distribution estimation problem, where we try to find a distribution that reproduces observed data. In addition, the entropy regularization has a very different interpretation. In Dyer et al. (2024), the entropy regularization “exhibits the trade-off between (a) the diversity of synthesized populations and scenarios and (b) the ability to identify the most extreme manifestations of the target scenario”. In our work, we show theoretically (Prop. 2.1) and empirically (Table 1, Figures 3,4,5) that no such trade-off is necessary for source distribution estimation.
>
> In summary, we believe that our work is a significant step upon previous work and hope that our response addresses the reviewer’s concern that our contribution is limited to the use of a sample-based loss.
>
>
> **Q1: "Is there anything that we lose by using something like the Sliced Wasserstein Distance instead of a likelihood-based discrepancy?"**: We thank the reviewer for their question. We believe that the main drawback of using sample-based distances is in the low sample limit, where the number of observations $N$ in the true data is very small. Likelihood-based objectives may perform better in this setting, as the value of the likelihood itself contains more information than the sample alone.
>
> We also evaluated Sourcerer’s performance in a low sample count regime ($N=100$ instead of $N=10000$) in a new experiment for the Two Moons task (Fig. R1a). We observe that our method, using the sample-based Sliced-Wasserstein distance, is still able to recover good sources in this case.
>
> **Q2: "Perhaps something additional to discuss would be limitations associated with simulators that involve discrete randomness"**: We thank the reviewer for pointing out this limitation. While we believe that an investigation of discrete simulators is outside the scope of this work, we agree that it merits acknowledgement in the limitations section.

---

> > ### Comment · Area_Chair_uKg1 · 2024-08-13
> >
> > Dear reviewer, please read the above rebuttal and evaluate whether it answers your concerns. If your evaluation remains unchanged, please at least acknowledge that you have read the author's response.

---

> > ### Comment · Reviewer_J2pY · 2024-08-13
> >
> > Thanks for your detailed response – I'm happy with all of your points.

---

> > > ### Author Response · Authors · 2024-08-14
> > >
> > > We thank the reviewer for their response.

---

### Official Review · Reviewer_DWPT · 2024-07-10

**Soundness:** 3
**Presentation:** 4
**Contribution:** 3
**Rating:** 7
**Confidence:** 3

**Summary:**

The authors propose to use the maximum entropy principle (possibly tempered with a prior) in order to reduce the ambiguity in solving the problem of source distribution estimation. They propose to use a sample-based technique that optimizes for the Sliced-Wasserstein distance measuring the discrepancy between the simulated data and the real data. They propose experiments on both synthetic and real-world models.

**Strengths:**

- The paper is extremely clear, concepts are presented consequentially and the method and experiments are easy to follow
- The method is simple, but principled and effective on both synthetic and realistic data
- The paper represents an interesting contribution to the field of source estimation methods.

**Weaknesses:**

I do not see any major weakness in the work. A minor one:

- W1: it would be interesting to see a more thorough study for models with even higher dimensionality, and how the proposed method's performance scales.

**Questions:**

- Q1: In some lines authors frame as the maximum entropy principle as something intrinsically beneficial. As they already discuss, this can be modulated by using a reference distribution. Maybe some statements that make the maximum entropy principle stand out as a sort of gold standard could be toned down a bit?
- Q2: Would it be possible for the authors to show what could happen with the choice of other or wrong distance types?
- Q3: An interesting case the authors could discuss is when multiple losses may be required for different subsets of parameters?

**Limitations:**

The authors already address the limitations, or the limitations are intrinsic to the field and cannot be attributed to the proposed methodology.

---

> ### Author Rebuttal · Authors · 2024-08-06
>
> We thank the reviewer for their positive assessment of our work. We address the reviewer’s questions below.
>
> **W1: "It would be interesting to see a more thorough study for models with even higher dimensionality"**: We thank the reviewer for their suggestion to investigate the performance of our approach in high-dimensional parameter spaces. To do so, we perform an additional experiment on the Gaussian Mixture benchmark task. The model is identical to the previous Gaussian Mixture model described in our benchmark task (Appendix 1.4), but now the dimension has been increased to 25. We again find sources that reproduce the data well, and that when we apply entropy regularization, we obtain higher entropy source distributions without any degradation to the pushforward accuracy (Fig. R1b). We also observe that the threshold value of $\lambda$, at which the pushforward C2ST degrades is now lower than in the 2-dimensional case. We believe that this is due to the entropy of the higher dimensional sources varying on a larger scale, thus restricting our method to smaller values of $\lambda$ to avoid the entropy term dominating the mismatch term.
>
> **Q1: "Maybe some statements that make the maximum entropy principle stand out as a sort of gold standard could be toned down a bit?"**: We agree with the reviewer that statements about maximum entropy being the "gold standard" need to be toned down, as this was not our intention. To clarify, we propose maximum entropy as one possible way to solve the ill-posedness of the source distribution estimation problem, which has a number of advantages such as uniqueness, and coverage of all feasible parameters for downstream inference tasks. These are not "intrinsically" optimal (and there might be specific cases where other regularizers would be more appropriate), and we will clarify the relevant statements in the revised version.
>
>
> **Q2: "Would it be possible for the authors to show what could happen with the choice of other or wrong distance types?"**: We thank the reviewer for their suggestion to use distance metrics other than the Sliced-Wasserstein distance (SWD). Therefore, we repeat our experiments on the benchmark tasks, replacing the SWD with MMD using an RBF kernel and the median distance heuristic for selecting the kernel length scale (Fig. R3). We observe that the results are comparable in terms to the SWD results presented in our work.
>
> The main considerations in choosing the SWD in our work were computational. We expect that any sample-based, differentiable, and computationally efficient distance metric to measure the mismatch in our objective function (Eqs. 3, 4) will provide reasonable performance. Our initial choice of using SWD was motivated by its simplicity and scalability - we did not need to choose kernels or other hyperparameters (except for the number of slices).
>
> **Q3: "An interesting case the authors could discuss is when multiple losses may be required for different subsets of parameters?"**: We strongly agree with the reviewer that there are applications where a general mismatch function such as the SWD might be insufficient to measure the distance between distributions, and some combination of distance functions would be required for different subsets of the data. For example, for simulators that produce multimodal outputs, the SWD may not be an appropriate metric. Instead, we could consider designing a distance custom distance metric that combines the distances over the different modalities of the data. We believe that this is outside the scope of the work, but believe that this is an important point to add to the discussion in our submission.

---

> > ### Comment · Reviewer_DWPT · 2024-08-10
> > **Response to rebuttal**
> >
> > I have read both the rebuttal and the other reviewers' concern. I do not think the lack of differentiability of the simulators is a concern, as indicated by the authors in the rebuttal. I also appreciate the comparison with NEB and additional experiments on convergence. Given the rebuttal addresses satisfactorily all the points I raised and I do not find the points raised by other reviewers concerning I maintain my positive assessment of the work.

---

> > > ### Author Response · Authors · 2024-08-14
> > >
> > > We thank the reviewer for their response.

---

### Official Review · Reviewer_swSP · 2024-07-14

**Soundness:** 3
**Presentation:** 3
**Contribution:** 2
**Rating:** 4
**Confidence:** 3

**Summary:**

The paper introduces “Sourcerer,” a new method for source distribution estimation, focusing on maximum entropy distributions to effectively handle ill-posed problems common in simulating scientific phenomena. This approach leverages the Sliced-Wasserstein distance for sample-based evaluation, offering a significant advantage for simulators with intractable likelihoods, and is demonstrated to recover high-entropy source distributions without sacrificing simulation fidelity across various tasks.

**Strengths:**

- Utilizing the Sliced-Wasserstein distance for evaluating distribution discrepancies enables the method to operate effectively with simulators that have intractable likelihoods.

- The approach is rigorously tested across multiple scenarios, including high-dimensional observation spaces and complex simulators. The results demonstrate that it not only maintains high fidelity in reproducing the empirical distributions but also achieves higher entropy in estimated source distributions compared to existing methods.

**Weaknesses:**

- The proposed method only applies to differential simulators. I know a lot of these problems in computational biology have non differentiable simulators, as well as the Ising models. This is the drawback of the proposed method.

- I also wish there is more theoretical results in terms of the statistical consistency and error bounds of using the sliced Wasserstein distance. Because the sliced Wasserstein distance is approximated via empirical measures. The estimation error is highly depending on the sample size.

- I'm not sure if the maximum entropy approach is aligned with the intuition. What if the source distribution is highly concentrated in an area with high prob mass? I know $\lambda$ can be tuned to control it but will it affect the original problem a lot?

**Questions:**

- what is the motivation of using the sliced Wasserstein distance? There are other types of distances can be used for this problem. For example, the maximum-mean discrepancy (MMD) and the kernelized Stein discrepancy (KSD). MMD and KSD also have their sliced versions.

- Is there any principle of how to choose the hyperparameter $\lambda$?

- I know the sliced Wasserstein distance is not really informative because it is estimated via samples thus the density information is a sort of missing. Is there any empirical result of the convergence speeds for the proposed method vs the benchmark NEB?

**Limitations:**

see the above.

---

> ### Author Rebuttal · Authors · 2024-08-06
>
> We thank the reviewer for their thorough feedback. The reviewer raises some questions and concerns about our approach, which we address below.
>
> **W1: "The proposed method only applies to differential simulators"**: In the case where the simulator is neither differentiable nor provides explicit likelihood evaluations, our method can still be used, but then requires training a surrogate model.  On our benchmark tasks (Table 1), we have shown that using a surrogate instead of the simulator led to comparable results in terms of simulation fidelity to data, and entropy. The results for the realistic Hodgkin-Huxley task (Fig. 5) were also obtained using a surrogate model. We emphasize that our method has less requirements than previous approaches: We do not require that the simulator have an explicit and differentiable likelihood, but only that the simulator be differentiable. Thus, the set of simulators to which we can apply our method _without_ training a surrogate is larger.
>
> **W2: "I also wish there is more theoretical results in terms of the statistical consistency and error bounds of using the sliced Wasserstein distance"**: The Sliced Wasserstein distance (SWD) is well established and its theoretical properties are actively studied. In particular, the sample complexity of the convergence rate has been studied in several existing works. For example, Nadjahi et al. (2020) showed that the finite sample estimate of the SWD between two distributions converges with rate $\sqrt{N}$ in the number of samples $N$.  Furthermore, Nadjahi et al. (2019) showed that generative models trained to minimize the SWD using finite sample size $N$ also converge to the true optimizer (in distribution) at rate $\sqrt{N}$. We will add a summary of these properties in our revisions.
>
> We agree with the reviewer that our benchmark tasks would benefit from exploring how our method performs in the low data limit. Therefore, we provide an additional empirical result by repeating the Two Moons benchmark task with a smaller observed dataset of only 100 samples (Fig. R1a). We observe similar behavior to the baseline case presented in our work for 10000 observations.
>
> [Nadjahi et al. (2020)] - Statistical and Topological Properties of Sliced Probability Divergences. NeurIPS, 2020.
> [Nadjahi et al. (2019)] - Asymptotic Guarantees for Learning Generative Models with the Sliced-Wasserstein Distance. NeurIPS, 2019.
>
> **W3: "I'm not sure if the maximum entropy approach is aligned with the intuition"**: We apologize for not making the intuition for using the maximum entropy principle sufficiently clear. We provide further clarification here and will reflect this in our revisions.
>
> The maximum entropy principle does not conflict with the case where “the source distribution is highly concentrated in an area with high probability mass”. If parameters outside this area are not consistent with the data observed, our method will result only in distributions that are concentrated in this area. We observe this empirically in the deterministic SIR experiment (Fig. 4), where the source distribution is concentrated and regularization does not lead to higher entropy sources.
>
> Instead, our use of the maximum entropy principle can be viewed as resolving the non-uniqueness of the source distribution estimation problem. According to the maximum entropy principle, given a choice between two source distributions _that are consistent with the data_, one should conservatively choose the source distribution with the higher entropy (which is more dispersed). This is often desirable when source estimation is used to learn prior distributions, as it ensures that a wide range of possible parameters is considered.
>
> **Q1: "What is the motivation of using the sliced Wasserstein distance?"**: We thank the reviewer for their question. We agree that other choices of sample-based, differentiable distances are possible. We now perform a new experiment on the benchmark tasks replacing SWD with MMD with an RBF kernel and the median distance heuristic for selecting the kernel length scale (Fig. R3). We observe that the results are comparable to the SWD results presented in our work in terms of simulation fidelity to data and entropy.
>
> The constraints on the distance function D are computational; D should be differentiable, fast to compute, and not require probability densities. Our new empirical results show that indeed other distances that satisfy these constraints lead to good performance of our method. Our initial choice of using SWD was motivated by its simplicity and scalability - we did not need to choose kernels or other hyperparameters (except for the number of slices).
>
> **Q2: "Is there any principle of how to choose the hyperparameter?"**: We empirically find that a small value of $\lambda$ (e.g. $\lambda=0.015$), together with our linear decay schedule, is sufficient to obtain substantially higher entropy source distributions in all our experiments. Thus, starting with a small $\lambda$ is likely to be sufficient. In addition, a run without any regularization can be performed to verify that the regularization does not negatively affect the quality of the estimated source. We agree with the reviewer that more robust methods for choosing $\lambda$ are an interesting question for future research, and will reflect this in our revisions.
>
> **Q3: "Is there any empirical result of the convergence speeds for the proposed method vs the benchmark NEB?"**: We thank the reviewer for their question, and agree that our work can benefit from explicitly comparing the empirical convergence speed of Sourcerer to the baseline. We provide an empirical measurement of the convergence times of our method (with and without regularization) as compared to NEB, measured on the benchmark tasks (Table R1). The source model architectures $q_\phi$ are the same for all methods, as reported in Table 1 of our original submission. Sourcerer is faster than NEB on all benchmark tasks.

---

> > ### Comment · Area_Chair_uKg1 · 2024-08-13
> >
> > Dear reviewer, please read the above rebuttal and evaluate whether it answers your concerns. If your evaluation remains unchanged, please at least acknowledge that you have read the author's response.

---

### Author Rebuttal · Authors · 2024-08-06

We would like to thank the reviewers for their detailed engagement and constructive and positive feedback. Our paper introduced Sourcerer, a method for estimating maximum-entropy source distributions with sample-based distances. Reviewers found our approach to be “well-motivated” (J2pY) and “simple, but principled and effective” (DWPT). They commended our evaluation “[on] a wide variety of benchmarks” (8Zuo) to be “extensive” (J2pY) and “rigorous” (swSP). Finally, all reviewers found the presentation of our work to be good or excellent, with DWPT calling it “extremely clear”.

We have responded to each of the reviewers in detail to clarify their questions and address their  concerns. We have performed several additional experiments (see supplement) to support our response. Below, we summarize the main points raised by the reviewers, in addition to the results of the new experiments. We hope that our responses address the reviewers’ concerns and allow them to recommend our work for acceptance.

**Clarifying the use of the maximum entropy principle (swSP, J2py, DWPT)**: The source distribution estimation problem is known to be ill-posed - we here proposed to use the maximum entropy principle to address this issue, and showed (Prop. 2.1) that the resulting source distribution is unique. Maximum entropy approaches are often motivated by the idea of “maximum ignorance” - i.e., finding a distribution that makes the least assumptions about the model while satisfying all the known information. A maximum entropy approach is also arguably desirable when source estimation is used to learn prior distributions, as it ensures that a wide range of possible parameters is considered. We showed empirically that our regularized approach is able to obtain a high entropy source distribution without sacrificing the quality of the match between simulations (from the estimated source) and observations.

Reviewer swSP raised a concern about the case where “the source distribution is highly concentrated in an area with high probability mass”.  However, if the parameters outside this area are not consistent with the observed data, our method will result only in distributions that are concentrated in this area. Nevertheless (and as with any regularization term), there are cases where other properties are desirable and where there might be more appropriate objectives to select the source distribution (as also pointed out by DWPT). We will provide a balanced discussion of when entropy regularization might be desirable.

**Clarifying the requirement of differentiable simulators (JA3k, swSP, J2py)**: Another shared concern was the requirement that the simulators be differentiable. We emphasize that this requirement is less restrictive than the ones from previous work (Vandegar et al. (2020)), which required the simulator _and_ its (log-) likelihood to be differentiable. Furthermore, if the simulator is not differentiable, or if computing its gradients is too expensive, our method can still be used. In this case it is necessary to train a surrogate model of the simulator. For example, in our manuscript, we train a surrogate model in the Hodgkin-Huxley experiment because some of the summary statistics used are not differentiable.

[Vandegar et al.] - Neural Empirical Bayes: Source distribution estimation and its applications to simulation-based inference. AISTATS, 2020.

## New Experiments

For new figures (Fig. Rx) and Table R1, please see the **PDF**.

**1. Estimating source distributions with higher dimensionality and less observations (DWPT, swSP)**: Reviewers asked for additional experiments to study the performance of our method in two cases. We now study the robustness of our method to a small number of observations in the dataset (swSP). We repeat the benchmark Two Moons task with $N=100$ observations, and find that our method still identifies high entropy source distributions that reproduce the observed dataset very well (Fig. R1a). We also add a new experiment studying the robustness of our method in the high-dimensional source distribution limit (DWPT). We repeat the Gaussian Mixture task with the source distribution dimension increased to $D=25$ (Fig. R1b), which is almost twice the dimensionality of the highest-dimensional source estimated in our original submission ($D=13$ in the Hodgkin-Huxley example). Again, we find that our method identifies a high entropy distribution with excellent match to the observed data.

**2. Additional comparison against NEB baseline (8Zuo, swSP, JA3k)**: Reviewers requested comparisons to baseline models in terms of convergence speed. We now show empirically that Sourcerer is significantly faster than the Neural Empirical Bayes (NEB) baseline across all benchmark tasks (Table R1 in supplement). In addition, we now also compare our results for the high-dimensional tasks of the SIR and LV models to the NEB baseline (Fig. R2) and show that Sourcerer performs significantly better on these tasks.

**3. Sourcerer with MMD (swSP, DWPT, JA3k)**: Reviewers asked whether the choice of the Sliced-Wasserstein distance (SWD) to measure the mismatch between simulations from our source model and the data was the only appropriate choice. We have clarified the constraints that our approach requires of the distance function to satisfy, namely that the distance must be sample-based, differentiable, and fast to evaluate. We chose to use the SWD because it meets all of these requirements and also requires few hyperparameter choices. However, other sample-based distance metrics are also possible: We have now performed additional experiments on our benchmark tasks, replacing the SWD with the Maximum Mean Discrepancy (MMD). We observe similar quantitative performance as with SWD (Fig. R3).

---

### Comment · Area_Chair_uKg1 · 2024-08-08

Dear authors and reviewers,

The authors-reviewers discussion period has now started.

@Reviewers: Please read the authors' response, ask any further questions you may have or at least acknowledge that you have read the response. Consider updating your review and your score when appropriate. Please try to limit borderline cases (scores 4 or 5) to a minimum. Ponder whether the community would benefit from the paper being published, in which case you should lean towards accepting it. If you believe the paper is not ready in its current form or won't be ready after the minor revisions proposed by the authors, then lean towards rejection.

@Authors: Please keep your answers as clear and concise as possible.

The AC

---

> ### Comment · Area_Chair_uKg1 · 2024-08-13
>
> Dear all,
>
> Thank you for your efforts so far in reviewing the paper and in answering reviewers' questions.
>
> The evaluation is currently borderline, with an average score of 5.8 and a spread from 4 to 7.
>
> The authors-reviewers discussion will close today on August 13. Please take this last opportunity to discuss the paper and the reviews and to reach a consensus, or at least to gather all the necessary information to make a fair and informed decision (in favor of acceptance or rejection).
>
> Thank you for your attention and cooperation.
> The AC

---

### Decision · Program_Chairs · 2024-09-25

**Decision:**

Accept (poster)

**Comment:**

Except for Reviewer swSP, all reviewers recommend acceptance (4-7-5-6-7). Reviewers all agree on the soundness of the approach, on the overall clarity of the presentation, as well as on the quality of the empirical evaluation. In my opinion, this contribution also makes a valuable step forward in the field; proposing to follow the maximum entropy principle when solving the source estimation problem is well-principled and the use of a sample-based loss does fit well with the simulation-based inference setting, even if the differentiability of the simulator is required.

The authors-reviewers discussion has been constructive and has led to a number of clarifications and new results, to the satisfaction of the reviewers. The concerns raised by Reviewer swSP are legitimate, but the authors have, in my opinion, provided convincing answers and new results to address them. Despite the lack of acknowledgment from the reviewer, I therefore consider that the major concerns have been addressed.

For all these reasons, I recommend acceptance. The authors are asked to implement the clarifications and add the new results discussed with the reviewers in the final version of the paper.